# CHARACTERISING THE INDUCTIVE BIASES OF NEURAL NETWORKS ON BOOLEAN DATA

## ABSTRACT

Deep neural networks are renowned for their ability to generalise well across diverse tasks, even when heavily overparameterized. Existing works offer only partial explanations (for example, the NTK-based task-model alignment explanation neglects feature learning). Here, we provide an end-to-end, analytically tractable case study that links a network's inductive prior, its training dynamics including feature learning, and its eventual generalisation. Specifically, we exploit the one-to-one correspondence between depth-2 discrete fully connected networks and disjunctive normal form (DNF) formulas by training on Boolean functions. Under a Monte Carlo learning algorithm, our model exhibits predictable training dynamics and the emergence of interpretable features. This framework allows us to trace, in detail, how inductive bias and feature formation drive generalisation.

## 1 INTRODUCTION

Deep neural networks have achieved remarkable success despite being vastly overparameterized (Brown et al., 2020; Jumper et al., 2021), challenging classical learning theory predictions that such models should overfit due to their capacity to learn highly complex functions (Zhang et al., 2016). As a result, much work has gone into studying the inductive biases of neural networks as a means for explaining their ability to generalise (Chizat & Bach, 2020; Belkin, 2021; Delétang et al., 2023). While several partial explanations exist, we lack a complete framework that connects architectural design, learning dynamics, feature emergence, and generalisation in an analytically tractable manner.

Understanding neural network generalisation can broadly be organised into three lines of work. *(i) Kernel-based theories* treat a network's training dynamics in the infinite-width limit as linearised gradient descent most prominently through the Neural Tangent Kernel (NTK) (Jacot et al., 2018; Ortiz-Jiménez et al., 2021). These analyses show that the network first fits functions aligned with the top eigenfunctions of the kernel (Bowman & Montufar, 2022). However, this framework cannot capture feature learning as weights do not evolve during training. *(ii) Finite-width feature-learning* analyses the training dynamics but mostly in mean-field settings or for linear neural networks for specific data distributions (Chizat et al., 2019; Mei et al., 2018; Dominé et al., 2025). Lastly, *(iii) Mechanistic interpretability* reveals meaningful structures in trained networks like feature detectors in vision models (Elhage et al., 2021) or induction heads in language models (Nanda et al., 2023). It identifies what representations emerge without explaining why or how these features develop through the interplay of architecture, data, and training dynamics. Despite deep neural networks' impressive generalisation in many domains and numerous insightful partial theories, there is still no end-to-end understanding of generalisation in neural networks from first principles.

### 1.1 OUR CONTRIBUTION

The aim of this paper is to introduce a toy model that lets us follow, step by step, how inductive bias (architecture and weight initialization), training dynamics and feature learning combine to yield generalisation. Concretely, we study *depth-2 discrete fully-connected networks* (DFCNs) on Boolean functions and show:

1. **Interpretable complexity measure:** We prove a one-to-one correspondence between DFCNs and disjunctive normal forms (DNFs) (Proposition 2.7). This allows us to translate geometric notions such as weight norm into a function-level complexity measure $K(f)$.

2. **Analytic characterisation of implicit bias:** Randomly initialised DFCNs induce a tractable prior $P(f)$ over Boolean functions. We derive bounds on $P(f)$ in terms of $K(f)$ that show strong simplicity bias in a range of function families (Table 2).

3. **Feature learning under a Bayesian lens:** Using both Metropolis sampling and a greedy stochastic gradient descent (SGD)-like algorithm, we demonstrate that generalisation correlates with the function's prior probability $P(f)$. Functions occupying larger volumes in parameter space (which have low DNF complexity) have low sample complexity, while those with small prior probability (like parity) are unlearnable – more data *harms* generalisation.

4. **Weight decay induces stronger simplicity bias and improves feature learning:** For DFCNs, $\ell_1$-regularisation translates into a simple multiplicative factor $e^{-\lambda K(f)}$ in the posterior equation 7. This lets us quantify how weight decay sharpens the native simplicity bias in $P(f)$. This improves generalisation on "easy" targets (small $K(f)$) but not on inherently complex ones (e.g. high-order parity). We also quantify how this optimiser-induced bias can lead to learning better features.

## 1.2 RELATED WORK

For a full discussion of related work refer to Appendix A. (Mingard et al., 2021; Smith & Le, 2017; Neal, 2012) argued that generalisation is best understood in a Bayesian framework, viewing SGD's convergence as approximating a Bayesian posterior. (Valle-Pérez et al., 2018; Mingard et al., 2025; 2019) empirically demonstrated that the prior over functions in randomly initialized neural networks has a strong simplicity bias towards Lemple-Ziv(LZ)-simple Boolean functions, see also e.g. (Palma et al., 2019; Teney et al., 2025b). For Boolean function learning, see also (Abbe et al., 2025; 2023).

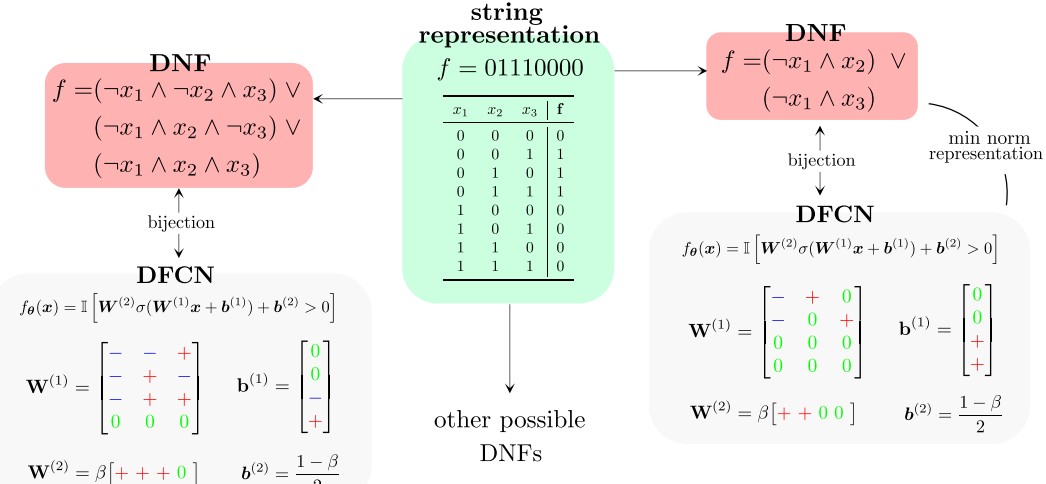

Figure 1: **Representing Boolean functions** Here we show the three ways of representing $f$. The **green** panel shows the string representation and truth table. The **left red** panel shows how we can extract the DNF representation from the truth table. The **right red** panel shows the minimum DNF representation of $f$ – when the complexity $K(f)$ is minimised. The **grey** panels show how we can represent $f$ by copying the clauses from the **red** panels into a DFCN (with '+'s meaning 1 and '-'s meaning $-1$). $W^{(1)}$ and $b^{(1)}$ are the weights and biases of the first layer. The combination of the ReLU activation and the bias term (set with Definition 2.6) ensures that each neuron's output is only 1 when the clause is True, and 0 otherwise. $W^{(2)}$ and $b^{(2)}$ act as the OR operators (plus a global function negation $\beta$). The example in the figure uses $\beta = 1$. Note that to guarantee full expressivity, $W^{(1)}$ has dimensions $(2^{n-1} \times n)$.

## 2 Preliminaries and intuition

In Section 2.1, we introduce Boolean functions along with two canonical representations: the string-representation and the DNF-representation. In Section 2.2 we introduce a novel DFCN-representation, which provides a one-to-one correspondence between Boolean functions and DFCNs.

### 2.1 Boolean functions and their disjunctive normal form

**Definition 2.1** (Boolean function $f$). For a fixed input dimension $n$, a Boolean function maps the $2^n$ vertices of the $n$-dimensional hypercube to $\{0, 1\}$

$$f : \{0, 1\}^n \to \{0, 1\}. \tag{1}$$

There are $2^n$ inputs $\boldsymbol{x} \in \{0, 1\}^n$ and 2 outputs $y \in \{0, 1\}$ and therefore $2^{2^n}$ possible functions.

**Definition 2.2** (string-representation). We define the *string*-representation of a function $f$ as an output string of 0s and 1s where the order is given by an ascending concatenated binary representation of the inputs. See Figure 1.

There is a second canonical way to represent a Boolean function: with the *disjunctive normal form* O'Donnell (2014). Any Boolean function with $n$ variables can be described by a truth table with $2^n$ rows (see Figure 1). Each row represents a complete assignment of the variables and therefore corresponds to a conjunction of literals (a clause). To obtain a symbolic description from this table, retain only the rows whose output entry is 1 and take the disjunction (logical OR) of their corresponding clauses. The resulting formula is the function's DNF, defined formally below.

**Definition 2.3** (Literal, clause, DNF). Let $\boldsymbol{x} \in \{0, 1\}^n$ denote a Boolean input vector. A *literal* is either a variable $x_i$ or its negation $\neg x_i$ for some index $i \in \{1, \ldots, n\}$. A *clause* $C$ is a conjunction (AND) of one or more literals: $C(\boldsymbol{x}) = \wedge_{i \in S} v_i$, $S \subseteq \{1, \ldots, n\}$, $v_i \in \{x_i, \neg x_i\}$. A Boolean function $f(\boldsymbol{x})$ can be described by a DNF with $t$ clauses if there exist clauses $C_1, ..., C_t$, and a global negation $\beta \in \{+1, -1\}$ such that

$$\Phi_f(\boldsymbol{x}) \; = \; \beta \big[ C_1(\boldsymbol{x}) \; \vee \; C_2(\boldsymbol{x}) \; \vee \; \cdots \vee \; C_t(\boldsymbol{x}) \big].$$

We note a slight abuse of notation in using $x$ to denote both input vectors and literals; the intended meaning should be clear from context.

**Definition 2.4** (Length of a DNF). For a DNF $\Phi_f$ whose $j$-th clause contains $k_j = |S_j|$ literals, its *length* is the total number of literals

$$L(\Phi_f) \; = \; \sum_{j=1}^{t} k_j. \tag{2}$$

When $\beta = -1$, we obtain the logical complement $\neg$DNF. With this additional global negation, every Boolean function admits a DNF with at most $2^{n-1}$ clauses (Mingard et al., 2019). This representation of length $L(\Phi_f) \le n \cdot 2^{n-1}$ is obtained by enumerating all truth-table rows whose output equals 1 (or 0 when $\beta = -1$, which may be required when $t > 2^{n-1}$); it is called the *canonical expansion* and, up to lexicographic ordering of the clauses, is unique (Kohavi & Jha, 2009; Quine, 1952). See Figure 1 for an example. In practice, however, a Boolean function often admits a far shorter DNF $\Phi_f$, and finding such a minimal representation is NP-hard (Allender et al., 2005). Conversely, the length can be artificially increased beyond $n \cdot 2^{n-1}$ by padding the formula with tautologically false clauses or duplicating existing ones, yielding DNFs of arbitrarily large length that still compute $f$. Because the raw length can therefore grow without bound, a meaningful complexity measure should refer to the *shortest* realisation of $L(\Phi_f)$ (see Appendix B.2 for further justification).

**Definition 2.5** (DNF complexity). We define the DNF complexity $K(f)$ as the shortest possible DNF expressing $f$,

$$K(f) \; = \; \min_{\Phi_f} L(\Phi_f). \tag{3}$$

## 2.2 DFCN–DNF CORRESPONDENCE

A central concept of this paper is to link the first two well-known representations of a Boolean function to one in terms of DFCNs with a fixed hidden layer width of $\alpha_w 2^{n-1}$, where $\alpha_w \in \mathbb{N}$. A DFCN with this structure is fully expressive (Mingard et al., 2019).

**Definition 2.6** (DFCN). A DFCN is a depth-two network, with ReLU activation $\sigma$, $f_\theta(\boldsymbol{x}) = \mathbb{1}\left[W^{(2)}\sigma\left(W^{(1)}\boldsymbol{x} + b^{(1)}\right) + b^{(2)} > 0\right]$, and the following weight structure:

| | | | |
|---|---|---|---|
| first-layer weights $W^{(1)}$ | $W^{(1)}_{ij}$ | $\in$ | $\{-1, 0, 1\}$ |
| first-layer bias $b^{(1)}$ | $b^{(1)}_i$ | $=$ | $1 - \sum_j [W^{(1)}_{ij} = +1]$ |
| second-layer weights $W^{(2)}$ | $W^{(2)}_i$ | $\in$ | $\beta \cdot \{0, 1\}$ |
| second-layer bias $b^{(2)}$ | $b^{(2)}$ | $=$ | $(1 - \beta)/2$ |
| global negation $\beta$ | $\beta$ | $\in$ | $\{-1, 1\}$ |

Table 1: DFCN construction.

The central reason we construct a discrete neural network with Definition 2.6 is that DFCNs are in one-to-one correspondence with DNF expressions.

**Proposition 2.7** (DNF-DFCN bijection). *For fixed $n$, there is a bijection between* (i) *parameter vectors $\theta$ satisfying the restrictions in Table 1, up to row permutations of $W^{(1)}$, and* (ii) *DNF formulas over $n$ variables, up to clause permutations and allowing for a global $\beta$ negation.*

*Proof.* For the proof see Appendix B.7. □

Figure 1 illustrates how the DFCN construction intuitively works. A $t$-clause DNF can represent any Boolean function with $t$ 1s. Each clause can be represented in a single row of the first layer of the DFCN, with the ReLU activation and correctly set bias term returning 1 if and only if the clause is satisfied. The second layer of 1s and 0s then acts as the OR operator (with a 0 ignoring the clause). The parameter $\beta$ is there to ensure symmetry between a function and its complement. As DFCNs and DNFs are in bijection, given a sufficient width, there also exist multiple DFCNs expressing the same Boolean function $f$. We define $\mathcal{W}^{(1)}_f$ as the set of all matrices $W^{(1)}$ that express $f$. We now relate the weight norm to the DNF complexity.

**Definition 2.8.** We set the norm of the weight matrices $(W^{(1)}, W^{(2)})$ to

$$\|W^{(k)}\|_1 = \sum_{ij} |W^{(k)}_{ij}| = \sum_{ij} \mathbb{1}[W^{(k)}_{ij} \neq 0], \tag{4}$$

$k \in \{1, 2\}$. $\|\theta\|_1 = \|W^{(1)}\|_1 + \|W^{(2)}\|_1$ denotes the overall norm of the DFCN.

$\|W^{(1)}\|_1$ corresponds to the number of non-zero entries in $W^{(1)}$, which is equivalent to the number of literals in the DNF representation, allowing us to relate this quantity to $K(f)$.

**Proposition 2.9.** *For $f$ represented as a DFCN $f_\theta$, the complexity is given by*

$$K(f) = \min_{W^{(1)} \in \mathcal{W}^{(1)}_f} \|W^{(1)}\|_1. \tag{5}$$

*Proof.* This directly follows from Proposition 2.7, see Appendix B.2 for more details. □

The "minimum representation" DFCN (right grey panel of Figure 1) has the lowest $K(f)$; equivalently, $\|W^{(1)}\|_1$ is minimized. This construction lets us directly link low weight norm to simple functions. Proposition 2.7 completes the picture of the following equivalent ways to express any Boolean function $f$:

- **String representation:** We can represent $f$ using a binary string of output values, ordered by input.
- **DNF representation:** We can represent $f$ using a DNF $\Phi_f$.
- **DFCN representation:** We can represent $f$ by a DFCN of width $\geq 2^{n-1}$.

## 3 UNTRAINED NEURAL NETWORKS

In this section we provide empirical results indicating that individual functions with low DNF complexity occupy a much larger fraction of parameter space than complex functions do. This bias towards simple functions influences training (Section 4).

### 3.1 A DFCN INDUCED PRIOR OVER BOOLEAN FUNCTIONS

Our goal is to understand which Boolean functions a *randomly initialised* DFCN is most likely to compute. Because, by Proposition 2.7, a depth-two DFCN is fully specified once its first-layer weight matrix $W^{(1)}$ is fixed, the most agnostic prior is to draw each entry of $W^{(1)}$ independently and uniformly from the ternary set $\{-1, 0, 1\}$. After $W^{(1)}$ is chosen we flip an unbiased coin for a global sign $\beta \in \{-1, 1\}$. If at least one hidden unit in a row in $W^{(1)}$ is non-zero, we set $W_i^{(2)} = \beta$, else $W_i^{(2)} = 0$. The two bias vectors are deterministic functions of $W^{(1)}$ and $\beta$, see Definition 2.6.

**Definition 3.1** (Prior probability). Let $\{\theta\}$ denote the finite set of weight vectors $\theta$ produced by the sampling procedure above, its size is $|\{\theta\}| = 2 \cdot 3^{n2^{n-1}}$. For a Boolean function $f$ we define

$$P(f) \;=\; \frac{\big|\{\theta \in \{\theta\} : f_\theta = f\}\big|}{|\{\theta\}|}, \tag{6}$$

i.e. the fraction of all admissible parameters that implement $f$.

Because each weight setting occupies a unit cell of equal size, $P(f)$ is proportional to the volume of weight space assigned to $f$. Boolean functions $f$ whose string representation can be implemented by many different weight configurations naturally claim a larger share of this volume.

### 3.2 SIMPLICITY BIAS IN $P(f)$

$P(f)$ has emerged as a strong predictor of generalisation (Mingard et al., 2021; Valle-Pérez & Louis, 2020). Under a Bayesian update with 01-likelihood on $m$ samples, the posterior weight of any interpolating function $f$ is exactly proportional to $P(f)$. Moreover, the PAC–Bayesian bound

$$\epsilon(f) \leq 1 - \exp\Big(\frac{-\ln P(f) + \ln(\delta/2m)}{m - 1}\Big)$$

implies that larger $P(f)$ yields tighter expected generalisation error.

Empirical studies show that the equivalent prior of continuous FCNs is heavily biased toward simple functions. Motivated by general arguments on overparameterised learners from (Dingle et al., 2018), Valle-Pérez et al. (2018) observed that $P(f) \lesssim 2^{-K_{\mathrm{LZ}}(f) + \mathcal{O}(1)}$, where $K_{\mathrm{LZ}}(f)$ is the Lempel-Ziv complexity of the string representation of $f$. While these findings suggested a fundamental connection between function probability and complexity, they relied on a complexity metric that lacks a connection to network architecture. In contrast, DNF complexity $K(f)$ provides a more interpretable measure with explicit ties to the network, as it directly counts the minimal literals needed to express the function (see Appendix B.3 for a further discussion on complexity metrics). This connection to the DFCN allows us to derive analytical bounds on $P(f)$ in the next section.

| Function class | Scaling | Complexity | $\log_2 (P)/K$ **ratio** |
|---|:---:|:---:|:---:|
| Constant | $0 \leq 1 - P(f) \leq 2^{-O(\alpha_w (4/3)^n)}$ | O(1) | $\alpha_w (4/3)^n$ |
| $t$-entropy $(t = O(n))$ | $2^{-O(\alpha_w t (4/3)^n)}$ | O(nt) | $\alpha_w (4/3)^n / n$ |
| $k$-parity | $2^{-\Theta(\alpha_w k 2^{n-1})}$ | $k2^{k-1}$ | $\alpha_w 2^{n-k}$ |

Table 2: Scaling laws in $P(f)$ as a function of complexity. We only show the leading order terms, valid when $\alpha_w \gg (3/4)^n$.

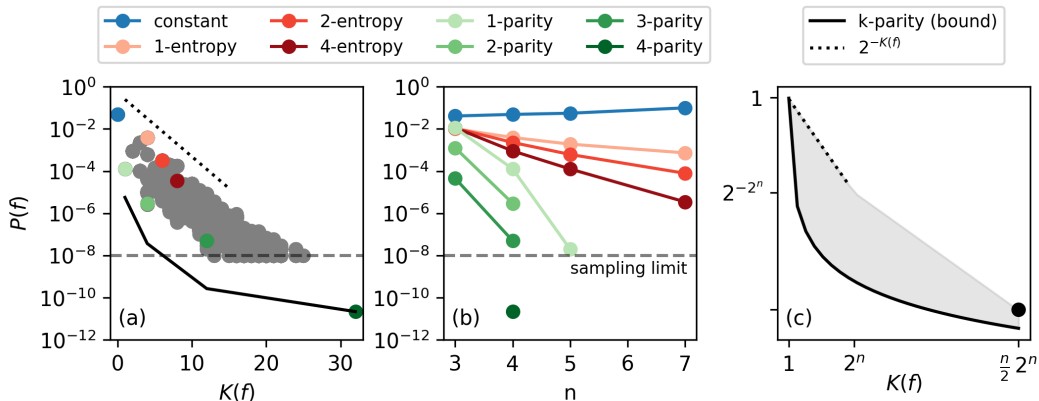

Figure 2: **Prior probability $P(f)$ vs. DNF complexity $K(f)$ for $n = 4$.** The hard cutoff at $P(f) = 10^{-8}$ reflects sampling constraints from $10^8$ parameter draws. **(a)** Each point represents a Boolean function, with constant functions (blue, $K = 0$) dominating the parameter space. Low-complexity functions occupy exponentially larger volumes, with $k$-parity (greens) suppressed compared to $t$-entropy (reds) of equal complexity. **(b)** Function probability scaling with input dimension $n$ shows $k$-parity probability decreasing much faster than $t$-entropy, matching theoretical bounds in Table 2. **(c)** Asymptotic bounds on $P(f)$ for large $n$ values. The black point is parity.

### 3.3 Understanding $P(f)$ v.s. $K(f)$

Figure 2(a) compares the empirical prior probability $P(f)$ obtained from $10^8$ i.i.d. parameter draws to the DNF complexity $K(f)$ for $n = 4$ (see Appendix C for full details). Only 631 of the total 65536 functions were not found, indicating that $P(f) \lesssim 10^{-8}$ for these functions. Each datapoint is a function. The minimum complexity constant function (blue) is the most frequent function, with the random $t$-entropy functions (reds) occupying the upper part of the envelope, and $k$-parity (greens) the lower. Figure 2(b) shows the dependence of $P(f)$ on $n$ for some function types. $k$-parity fall much faster with $n$ than the $t$-entropy with $t = 1, 2, 4$. Can we predict $P(f)$ v.s. $K(f)$ at large $n$?

**Theorem 3.2.** *We require $\alpha_w \geq 1$ to satisfy full expressivity. To leading order, for the three function classes defined in this section, $P(f)$ scales as Table 2.*

*Proof.* See Appendices D.2 to D.4 for bounds on the constant function $t$-entropy and $k$-parity. $\square$

We study three canonical families of functions (full descriptions in Appendix D). **Constant functions:** $f$ outputs the same label on every input – $K(f) = 0$ (minimum representation has $W^{(1)} = 0$). $k$-**parity:** Sparse parity on the first $k$ bits – $K(f) = k2^{k-1}$. $t$-**entropy:** Exactly $t$ ones and $2^n - t$ zeros – $K(f) \leq n \min(t, 2^n - t)$. These classes of functions allow us to explore a broad range of complexities. We summarise the most relevant bounds and scaling laws in Table 2 (for a full discussion of bounds and assumptions, see Appendix D).

Valle-Pérez et al. (2018) predicted that $\log_2(P(f)) \leq -aK + b$ for some constants $a, b$ (independent of $K$), and empirically observed that for a large class of functions, the bound was an equality. Table 2 provides theoretical results showing that for the typical $t$-entropy function, this scaling is a function not of $K(f)$, but $\alpha_w t (4/3)^n$, multiplying the complexity term by $\alpha_w (4/3)^n / n$. However, for $k$-parity, $P(f)$ scales as $\alpha_w k 2^{n-1}$, suppressing the complexity term by a significantly larger factor, $\alpha_w 2^{n-k}$ (which is dependent on complexity). This extra suppression predicts that $P(f)$ will occupy a wide envelope – with some functions of low complexity but also low probability (a concrete example of predictions are presented in (Dingle et al., 2020)). This is visualised in Figure 2(c).

## 4 Trained neural networks

We use $n = 7$. All models are trained on random subsets $S \subset \{0, 1\}^n$ of size $m \in \{16, 32, 64, 96\}$, where the rest of the set $\{0, 1\}^n$ is used as a test set. Results are averaged over ten independent draws. All runs use the DFCN of Definition 2.6 with $\alpha_w = 2$ to ensure overparameterisation.

## 4.1 Training algorithms

As DFCNs have a discrete weight space, they cannot be straightforwardly trained using SGD. While SGD variants adapted to discrete weights exist (Hubara et al., 2016), we instead employ a fully Bayesian MCMC algorithm, due to its easier interpretability, and an oracle algorithm as described below. We also implement a steepest descent random search algorithm. See Appendix E for full details.

**Metropolis–Hastings (Alg. 1).** A Metropolis-Hastings algorithm based on the acceptance probability $\alpha = \min\{1, e^{-\kappa\Delta\mathcal{L}(\theta)}e^{-\lambda\Delta\|\theta\|_1}\}$, where $\Delta$ denotes the difference between steps, $\mathcal{L}$ is the MSE error, $\kappa$ is an inverse-temperature hyperparameter and $\lambda$ is the weight–decay coefficient.

**Min norm oracle (Alg. 2).** This is an oracle that returns the minimal complexity DNF compatible with the training set $S$ obtained by exhaustive search.

**Greedy SGD–like (Alg. 3).** This performs greedy local optimisation. At every step $\theta_t$, we evaluate the loss of every possible neighbour of $(W^{(1)}, W^{(2)})$ with Hamming distance one to $\theta_{t-1}$. We find the set of neighbours which maximally improve the batch error (minimising the loss), picking uniformly from the lowest-norm neighbours with probability $p$, otherwise picking uniformly from the entire set with probability $1 - p$. The hyperparameter $p$ acts like a weight decay parameter: larger $p$ results in a larger bias towards minimum norm functions and hence towards functions of small $K(f)$.

## 4.2 Weight decay adds an additional bias in the posterior

For Algorithm 1 we choose $\kappa = 1000$, which approximates the likelihood term $e^{-\kappa\Delta\mathcal{L}(\theta)}$ in the MCMC sampling as a 01-likelihood $\mathbb{1}[f_\theta(S) = f^*]$. With the uniform prior over $\theta$ defined in Definition 3.1, the posterior over Boolean functions $f$ is obtained by marginalising:

$$P_\lambda(f \mid S) = \frac{\sum_{\{\theta: f_\theta = f\}}\mathbb{1}[f_\theta(S) = f^*]e^{-\lambda\|\theta\|_1}P(\theta)}{\sum_\theta \mathbb{1}[f_\theta(S) = f^*]e^{-\lambda\|\theta\|_1}P(\theta)} \simeq \frac{e^{-\lambda K(f)}P_{\lambda=0}(f \mid S)}{\mathbb{E}_{f\sim P_{\lambda=0}(\cdot|S)}[e^{-\lambda K(f)}]},$$

where $P_{\lambda=0}(f \mid S)$ is just the posterior induced by a 01-likelihood, $P(f \mid S) \propto \mathbb{1}[f_\theta(S) = f^*]P(f)$. We have assumed that $\sum_{\{\theta: f_\theta = f\}} e^{-\lambda\|\theta\|_1}$ is dominated by the smallest attainable norm $\|\theta\|_1$ for a given $f$, and that $\|\theta\|_1 \simeq \|W^{(1)}\|_1$, which gets more accurate for larger $n$ since the parameter space is largely dominated by $W^{(1)}$. As the norm is directly related to the complexity (Proposition 2.9), weight decay approximately acts as a multiplicative bias $e^{-\lambda K(f)}$ that further sharpens the simplicity bias in $P(f)$.

The results of training DFCNs with Algorithm 1 are shown in Figure 3. Figure 3(a) shows the inductive bias of the network: higher complexity functions are harder for the network to learn due to the large bias towards low complexity functions. Hence, at zero training error, the network only achieves good test accuracy for simple target functions. Figures 3(b) and (c) show heatmaps of $W^{(1)}$ during training at different test accuracy checkpoints for a 4-parity target function. We see that for no weight decay ($\lambda = 0$), the network trains towards a simple representation of the target function (resulting in a sparse $W^{(1)}$ with many zeros), but adding weight decay ($\lambda = 0.01$) greatly improves the generalisation of the network by further increasing the bias towards low complexity functions. At 100% test accuracy (Figure 3(c)), we see that the network has learned the exact minimal DNF representation – i.e. optimal features – of the target function, with 4-parity requiring eight clauses in its minimal DNF.

This gives an intuitive explanation for the general empirical observation that weight-decay improves test performance and the general hypothesis that flatter minima (higher volume or prior weight $P(f)$) generalise better (He et al., 2020; Li et al., 2018; Tessier et al., 2022).

## 4.3 The special case of parity (Figure 4)

**MCMC without weight-decay ($\lambda = 0$).** As $\kappa \gg 1$, the posterior is dominated by the prior $P_{\lambda=0}$. Simple parities ($k \leq 4$) are eventually learned, while $k = 6, 7$ never exceed chance level despite more data. Their posterior mass is simply too small for the algorithm to find them within the allotted budget,

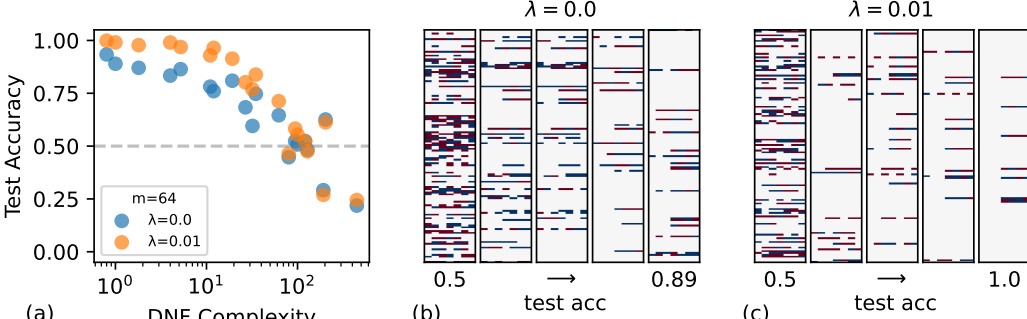

(a)  (b)  (c)

Figure 3: **Inductive biases of trained DFCNs** ($n = 7$) **with Algorithm 1** **(a)** shows how the inductive bias of DFCNs towards lower complexity functions allows them to find such functions more easily than higher complexity functions. It also shows that weight decay increases these biases, being able to achieve 100% test accuracy on some functions. See Appendix E for a list of the functions used. **(b)** and **(c)** show heatmaps of $W^{(1)}$ during training at different test accuracy checkpoints with a target function of 4-parity for no weight decay ($\lambda = 0$) and with weight decay ($\lambda = 0.01$), respectively. Both panels show how the network learns simple representations of the target function, with weight decay managing to learning the exact DNF representation.

mirroring the extreme rarity observed in Figure 2. Weight norms stay high and almost $k$-independent. See Appendix B.6 for a discussion on why more data actually harms generalisation for high-parity target functions.

**MCMC with weight-decay** ($\lambda > 0$). Penalising the norm dramatically improves generalisation when a low-norm representation exists. For $k = 1$ the network reaches perfect accuracy after $m = 64$ examples and its weight norm drops to a much lower value than without weight-decay. These performance gains from weight-decay persist up to $k = 4$. Beyond this the minimum representation is so large that the decay term can no longer offset the small prior probability equation 7.

**Min norm Oracle.** The oracle provides the Bayes-optimal accuracy achievable if the learner always selects the lowest-norm DNF that interpolates $S$. The accuracy gap between the weight-decay sampler and the oracle is small for $k \leq 4$, demonstrating that the algorithm actually discovers the minimal complexity DNF in practice even though it is capable of expressing much more complicated DNFs for the given string representation.

**Greedy SGD-like.** We train on parity as well as the other function families with the greedy SGD-like algorithm, see Fig. 12 in the Appendix. The learning curves look qualitatively similar to MCMC, showing that SGD behaves Bayesian in our DFCN setting, as also claimed in (Mingard et al., 2021).

## 5 DISCUSSION

State-of-the-art neural networks are so large and the data they ingest so heterogeneous that we can usually only fully understand curated sub-problems – for instance, modular arithmetic (Nanda et al., 2023). To make causal statements about representation learning we need a small, exactly-solvable test-bed in which (i) the target function's complexity is tunable, model-independent and human-interpretable, (ii) learned representations live in an easily interpretable space and (iii) the inductive bias and learned functions can be precisely understood.

Our DFCN offers precisely that by representing a DNF. The complexity $K(f)$ of the learning problem is controlled directly by the number and size of clauses of the target function $f$. $K(f)$ is therefore simultaneously meaningful for the data, the hypothesis class, and the parameters. Furthermore, the one-to-one mapping between weights and logical clauses lets us derive analytic expressions for the prior $P(f)$, turning qualitative notions of "simplicity bias" from (Valle-Pérez et al., 2018) into quantitative, testable predictions for generalisation. In this sense, the DFCN plays the same role in deep-learning theory that the Ising model plays in statistical physics: it is the minimal, exactly computable system that still exhibits the phenomena we care about. Sliding along the complexity

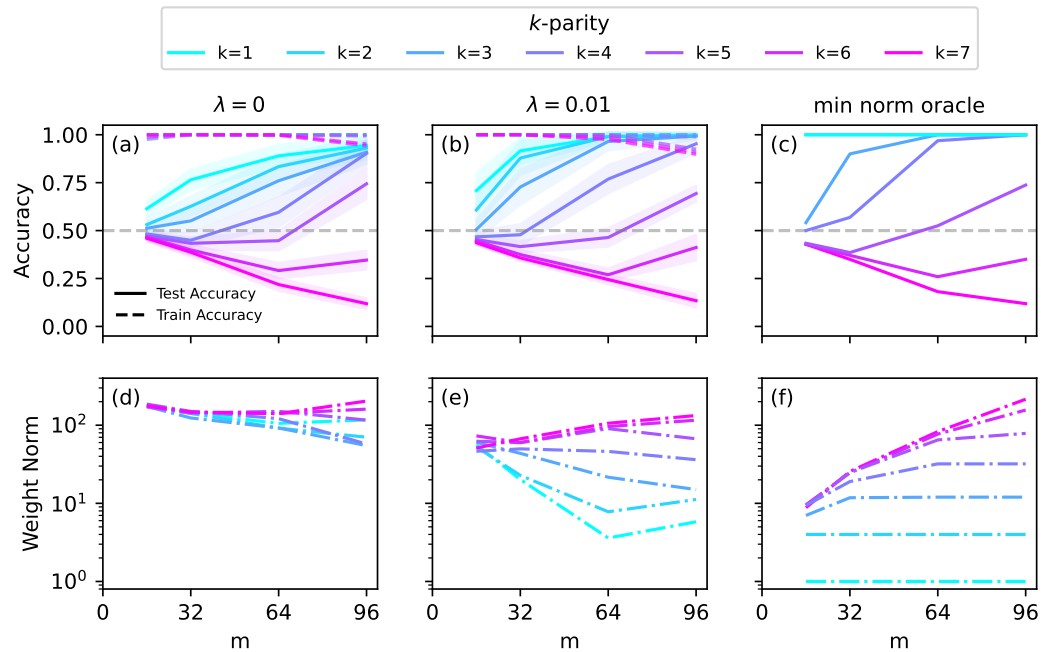

Figure 4: **Training statistics for $k$-parity target functions (a)**, **(b)** and **(c)** show training and test accuracies for various $k$-parity target functions for the MCMC algorithm (Algorithm 1) without weight decay ($\lambda = 0$), with weight decay ($\lambda = 0.01$) and an oracle algorithm (Algorithm 2), respectively. Weight decay improves test accuracy for all $k < 5$ functions. For 7-parity (the most complex function for an $n = 7$ input DFCN), the model is strongly biased against this function; the more data we give it, the worse the test accuracy will be. **(d)**, **(e)** and **(f)** show the weight norms for each of the training algorithms, clearly showing that weight decay greatly lowers the norm compared to no weight decay.

axis within a single framework demonstrates how complexity, inductive bias, training dynamics, and generalisation interact.

Specifically, we demonstrate analytically that DFCNs with uniform sampled weights (i) induce a strong simplicity bias in the distribution over Boolean function $P(f)$; (ii) this bias in the prior directly determines generalisation with a Bayesian learning algorithm, as best seen for high-parity Boolean functions where the bias against complex functions is so strong that it cannot overcome the prior, even with additional data points; and (iii) weight decay amplifies this simplicity bias, learning the minimum representation, inducing feature learning. This explains why it improves performance on simple target functions but not on inherently complex ones.

**Limitations.** Our work focuses on discrete networks with Boolean inputs, which provides analytical tractability but leaves a gap between our theory and typical continuous deep learning applications. The sampling algorithms become computationally intractable for large $n$, even though this is the interesting regime concerning the bounds in Table 2. Furthermore, our training algorithms do not capture all properties of continuous optimisation with SGD or other optimisers like Adam.

**Future directions.** One strength of this model is its ability to directly study the effect of optimiser hyperparameters. Exploiting this to study other phenomena observed in continuous neural networks such as grokking and neural collapse, via tuning of the weight decay parameter or increasing the width of the DFCN, would be interesting to explore. The most difficult task would be to develop suitable and interpretable optimiser measures that allow for bounds on the prior $P(f)$. This research direction should also better understand how different optimisers and training schemes influence the posterior, or whether a Bayesian formulation is even possible at all.

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

**LLM usage**  Large language models were used to aid in checking proofs and proofreading the text, but did not play a significant role in research ideation, retrieval or discovery.

# A  A BRIEF REVIEW OF GENERALISATION IN NEURAL NETWORKS

## A.1  EMPIRICAL STUDIES OF SIMPLICITY BIAS IN NEURAL NETWORKS

An increasing amount of work shows that randomly initialised standard neural network architectures possess an implicit "simplicity bias": that is, they prefer to learn functions that are algorithmically simple or low in complexity far more frequently than complex ones.

**Bias toward low-complexity functions in untrained neural networks**  Valle-Pérez et al. (2018) first quantified the bias of fully-connected neural networks on Boolean data by sampling functions from randomly initialised fully-connected networks and measuring their complexity via Lempel-Ziv compression as a proxy for Kolmogorov complexity. Using a bound derived from algorithmic information theory (Dingle et al., 2018), they found that the upper bound, $P(f) \leq 2^{-K_{LZ}(f)+\mathcal{O}(1)}$, was tight for small FCNs on boolean data ($K_{LZ}(f)$ is the Lempel-Ziv complexity of $f$, explained in Appendix B.1). This means simple functions are exponentially more likely to be realised by a random network than ones with a high Lempel-Ziv complexity (Dingle et al., 2024). Mingard et al. (2019) built on this observation, proving that, at initialisation, perceptrons are biased towards Boolean functions with low output entropy, meaning functions that output mostly 0's or 1's (which tend to be simpler) are favoured. This bias closely follows a Zipf-law or power-law distribution in function probability, indicating a heavy skew toward a few simple functions (see also Valle-Pérez et al. (2018)).

**Inductive bias of different architectures**  Different architectures, initialisation schemes and activation functions lead to different inductive biases in randomly initialised neural networks (Teney et al., 2025a). Schoenholz et al. (2017) showed a phase transition from ordered to chaotic information propagation in randomly initialised neural networks, depending on the variance of the random Gaussian weights. Teney et al. (2024) systematically examined random untrained networks with various activation functions, measuring their complexity in terms of their Fourier spectrum, polynomial expansion and (LZ) compressibility. They showed that standard multilayer perceptrons with ReLU or GELU activations strongly prefer low-frequency functions – effectively smooth input-output mappings – across different depths and weight scales. The opposite is true for Gaussian or sinusoidal activation functions, producing a very different prior in function space, where the inductive bias prefers higher-frequency (generally more complex) functions. See Yang & Salman (2020) for a similar kernel-based analysis.

**Inductive bias and real world data**  Different neural networks have different kinds of inductive biases. How much these biases improve or worsen generalisation depends on the structure of the target function. A common example is training an FCN and a CNN on an image dataset like CIFAR10: both will perform better than random, but the CNN will reach a significantly higher test accuracy. Convolutional networks trained on images have been found to latch onto simple, low-level cues (like textures or colours) rather than more complex global structures. Geirhos et al. (2018) showed that ImageNet-trained CNNs are strongly biased toward texture recognition rather than object shape. When presented with images where texture and shape cause conflict, CNNs predominantly follow the texture. Texture cues are, in a sense "simpler" or more immediate statistical features of images (requiring only local filtering), whereas global shape integration is more complex. This bias towards easy-to-pick-up features can hurt robustness, but it is an instance of the network's preference for a simple explanation of the data (here, classifying by texture) when one exists. See Fel et al. (2024) for a further discussion of the complexity of features in CNNs and how they arise during training.

Another line of work has examined transformer architectures for algorithmic or logical tasks. Bhattamishra et al. (2022) found that Transformers have a simplicity bias analogous to deep networks: they more readily learn low-sensitivity (sparse) Boolean functions than complex ones. For instance, a Transformer trained on a Boolean function that depends only on a small subset of input bits (with the rest being irrelevant noise) will generalise well, whereas learning a highly "entangled" function like parity (which depends on all bits) is notably difficult without special measures. This suggests the inductive biases of modern sequence models also favour functions with simple structures (e.g. ones

that can be decomposed into a few salient features or rules), even if the models in principle have the capacity to implement very complex mappings Fel et al. (2024).

Inductive bias only improves the performance of the model when it aligns with the target. Having well-performing models relies on real-world data being highly structured. Indeed, Goldblum et al. (2023) argue that this bias is one key reason we can have "general-purpose" models: real-world tasks themselves produce data that are far from fully random and instead have low underlying complexity, benefiting neural networks, which innately favour such low-complexity patterns. In their experiments, architectures specialised for one domain can often compress or model data from another domain if it shares a low-complexity structure, and even large pretrained language models with random weights preferentially generate low-complexity (compressible) sequences rather than arbitrary complex ones (Goldblum et al., 2023). This surprising observation – that an untrained GPT-style model already favours simplistic output patterns – reinforces that a great deal of inductive bias comes from architecture alone, not just from gradient descent (Teney et al., 2024).

**Bias towards low-complexity functions after training**   (Zhang et al., 2016) showed that neural networks are expressive enough to fit randomly labelled data. This raised the question of why neural networks trained on non-random labels generalise at all, since they would be perfectly capable of interpolating the training data while having random chance accuracy on the test data. This highlights that, beyond random initialisation, the neural network training process itself introduces an inductive bias. Mingard et al. (2021) argue that the posterior distribution over functions retains a simplicity bias – among all functions consistent with the training set, SGD-trained networks tend to land on ones of relatively low complexity. They offer empirical evidence on non-Boolean data that SGD is more likely to learn functions with larger $P(f)$ (see also (Naveh et al., 2021) for a theoretical explanation using infinitesimal GD with weight decay). Kalimeris et al. (2019) provided complementary evidence by examining training dynamics: they observed that SGD learns functions of increasing complexity over time, effectively learning a simple, approximately linear decision boundary in the early epochs and then gradually fitting more complex aspects of the target function in later epochs Early in training, almost all of the network's performance can be attributed to a "simple classifier" component, and only with more iterations does the model incorporate higher-order or more complex features. See Appendix A.3 for the kernel explanation of this phenomenon (or see Canatar et al. (2021)).

A.2   MECHANISTIC INTERPRETABILITY

While the studies discussed in Appendix A.1 treat neural networks mostly as black-box functions, a complementary line of research asks how exactly neural networks internally represent data – in effect, opening the black box to find mechanistic explanations for a network's behaviour. Mechanistic interpretability seeks to reverse-engineer the specific circuits and algorithms encoded in a trained network's weights. The goal is to move beyond coarse measures (like complexity or norm-based capacity) and identify neurons, attention heads, layers, or combinations thereof that correspond to meaningful functions or subroutines within the network.

**Interpreting CNNs – Circuits and feature visualisation**   In CNNs, interpretability research has progressed from identifying individual neurons that detect human-interpretable concepts to mapping out multi-neuron interactions, or circuits, that represent higher-level features. Early work used feature visualisation to synthesise an input image that maximally activates a given neuron or layer, revealing the feature that neuron represents (e.g. a neuron might fire for textures like "striped pattern" or specific objects like "dog faces").

In a series of articles (see e.g. (Olah et al., 2020; Elhage et al., 2021)), Olah et al. demonstrated that CNNS learn circuits for meaningful visual concepts. One notable example is a "curve detector" circuit (Cammarata et al., 2020): lower-layer neurons detect short curves at various positions; a mid-layer neuron sums these to detect longer curves; that neuron in turn feeds into a higher-layer neuron that detects round objects (like wheels or pupils), together forming a hierarchical circuit for round shapes. Such precise visual explanations illustrate how a network builds complex features (like detecting an animal face) by composing simpler ones (edges, textures, etc.). However, purely correlational methods face the challenge of polysemantic neurons (neurons that activate on multiple features).

Bozoukov (2025) used sparse autoencoders on InceptionV1's mixed layers to isolate more interpretable "feature vectors", which enabled tracing connections between features across layers to reconstruct circuits. They uncovered branch-specific circuits in the network – e.g. a chain of features detecting animal faces: early layers detect oriented animal parts (faces facing left or right), a next-layer feature combines them into a generic animal face detector, which then branches into specialised detectors for dog faces and dog legs in later layers. This kind of mechanistic story, where one can follow the activation flow through a sequence of feature detectors, represents a state-of-the-art understanding of how CNNs implement complex visual recognition. It provides a satisfying mechanistic explanation (complete with visual evidence) for tasks that the network learned – essentially reverse-engineering a portion of the network's computation graph in human-understandable terms.

**Transformer models – Induction heads and algorithms**   In Transformers, recent work has identified small-scale circuits inside large models that correspond to specific algorithms the model has learned. A prime example is the discovery of induction heads in Transformers. Induction heads are particular attention heads that implement a simple copy-and-paste algorithm: given a sequence pattern "[A][B] ... [A]", an induction head learns to attend from the second "[A]" back to the token "[B]" that followed the first "[A]", effectively retrieving "B" as the predicted continuation. Olsson et al. (2022) provided multiple lines of evidence that induction heads are the mechanistic basis of in-context learning in Transformers. They observed that at a certain training stage, models undergo a sudden jump in their ability to do in-context prediction of sequences (for example, continue a list in the style of the prompt), and this coincides with the emergence of one or two attention heads that reliably implement the above copy mechanism. By ablating those heads, the in-context learning ability drops, confirming a causal role. This finding is remarkable because it isolates a transparent algorithm within the black-box: the model learns to implement a memory lookup and retrieval operation entirely through a couple of attention heads (which are simple matrix-weighted operations). It's a rare case where one can point to specific weights in a large language model and say "this is performing task X via mechanism Y." Subsequent work has extended this analysis to larger models and more complex behaviours. For instance, Ren et al. (2024) identified "semantic induction heads" that not only copy tokens but do so in a way that respects word semantics (copying the next word of a repeated sequence even with intervening synonyms), showing the versatility of these circuits.

Beyond induction heads, interpretability researchers have attempted to reverse-engineer entire algorithms learned by Transformers on small tasks. For example, Li et al. (2024) fully explained how a tiny Transformer performs modular arithmetic. Nanda et al. (2023) studied grokking in transformers performing modular arithmetic, by reverse-engineering the algorithm the network learned for modular addition. They discovered that the 1-layer Transformer had learned to implement addition by internally converting numbers to a discrete Fourier representation (essentially representing integers as complex phases on a circle) and performing rotations. Accordingly, they defined progress measures for each component of the algorithm (e.g. how well the Fourier conversion sub-circuit was formed) and tracked them during training. They found that training proceeded in three phases: (1) memorization – the model first purely memorizes many training examples, (2) circuit formation – gradually the Fourier addition circuit emerges and gains strength, and (3) cleanup – finally the model prunes away the now-unneeded memorized solutions, relying purely on the general algorithm. What appeared as a sudden "grokking" jump in test accuracy was explained mechanistically as the point when the algorithmic circuit surpassed memorisation in importance.

## A.3 Kernel Methods and Spectral Perspectives on Generalization

Another avenue to understand inductive bias and generalisation is through the lens of kernel methods and spectral analysis. In the infinite-width limit (given the correct parameterisation of the neural network (Everett et al., 2024)), training the neural network with SGD corresponds to kernel regression with the so-called Neural Tangent Kernel (NTK) (Jacot et al., 2018; Lee et al., 2020). Analysing the eigenfunctions and eigenvalues of these kernels gives insight into what functions the network can learn easily – revealing the network's bias in function space in more mathematical terms.

**Spectral bias on the hypersphere**   For fully-connected ReLU networks, there are several works that provide a full description of the NTK eigenbasis. For fully-connected ReLU networks with input drawn from the $d$-dimensional hypersphere, the NTK is a dot-product kernel. Its eigenfunctions are the spherical harmonics on the hypersphere. The symmetry of the architecture together with the data

symmetry leads to a polynomial decay in the eigenfunction $k^d$ with the degree $k$ of the spherical harmonics (Basri et al., 2020; Geifman et al., 2020).

This means the kernel deems low-frequency spherical harmonics as "important" functions (high eigenvalue), and high-frequency as "hard" functions (low eigenvalue). Intuitively, this quantifies the simplicity bias of the network in a basis of functions: smooth, low-order functions lie in top-eigenvalue eigenspaces, whereas highly oscillatory or complex dependences lie in low-eigenvalue eigenspaces. A direct consequence is a spectral bias in learning. When one trains a neural network (in the kernel regime) or does kernel regression on data, the component of the target function along high-eigenvalue eigenfunctions is learned first and with few samples, while components along low-eigenvalue eigenfunctions require many more samples to fit.

Empirical studies have demonstrated this frequency bias: for instance, when fitting a target function on the unit circle that is a mixture of sines and cosines, neural nets quickly learn the low-frequency modes and only later fit the high-frequency components (Simon et al., 2021). Rahaman et al. (2019) showed that standard deep networks trained on regression tasks exhibit a Fourier frequency bias: the error in fitting different Fourier modes is controlled by the mode frequency, with low-frequency signals learned much faster (Canatar et al., 2021).

**Spectral bias on the hypercube**   These arguments transfer to neural networks trained with data from the hypercube. The NTK eigenfunctions on the hypercube are parity functions on subsets of $k$ input bits, and eigenvalues decrease as $k$ grows. Thus, a network can learn any function that is, say, an XOR of a few bits relatively easily (those correspond to eigenfunctions with high eigenvalue), but learning the parity of all $n$ bits (the hardest, $k = n$ eigenfunction) is extremely slow and sample-inefficient – essentially requiring memorization since that function lies in a low-eigenvalue subspace (Simon et al., 2021). This quantitatively explains why neural nets struggle with high-order parity (a well-known "hard" function class in theory) unless aided by exponential data or special architectural tweaks (Abbe et al., 2025): the inductive bias is simply misaligned with that function. On the other hand, a function like a single-bit identity or a simple conjunction of a few bits has most of its variance in low-order parity components and is learned readily. These insights bridge the gap between the empirical simplicity bias and a theoretical characterisation: networks have an implicit bias toward functions expressible by low-degree polynomials or low-frequency Fourier components, which are precisely the "simple" patterns in the input space.

The spectral bias argument can be extended to other architectures like CNNs (Geifman et al., 2022).

**Kernels do not do feature learning**   There is a rich literature on the sample complexity for kernels which provides a full theory of generalisation for kernels and hence infinitely wide neural networks (Cohen et al., 2021). However, there are several known datasets/target functions where neural networks strongly outperform kernels in terms of their sample complexity (Abbe et al., 2024; Refinetti et al., 2021; Ghorbani et al., 2020; Donhauser et al., 2021). I.e., neural networks learn the target function with a much lower number of datapoints. This is generally attributed to feature learning, the ability of the neural network to adapt its hidden representation (Bordelon et al., 2024). This part of deep learning cannot be understood through a kernel perspective.

## B   NOTES ON COMPLEXITY MEASURES

In this section, we will discuss DNF complexity $K(f)$ in more detail, and the alternative complexity measures. We will begin with Lempel-Ziv complexity, used on the string representation by (Valle-Pérez et al., 2018) in Appendix B.1. We then discuss the relation between $K(f)$ and the weight norm of the DFCN in Appendix B.2. In Appendices B.3 and B.4 we discuss properties of "good" complexity measures, and introduce alternatives (e.g. complexity equal to the number of clauses). In Appendix B.6 we explain why parity generalises so badly (specifically, why adding more data makes training accuracy decrease), and in Appendix B.7 prove the theorems relating DFCNs and DNFs stated in the main text.

## B.1 A PRIMER ON LEMPEL-ZIV COMPLEXITY

The Lempel–Ziv parsing (Lempel & Ziv, 1976) algorithm proceeds by scanning a finite string $x$ over any alphabet from left to right, maintaining a dictionary of distinct substrings (or "words") observed so far. Whenever the algorithm encounters a new substring that is not already in the dictionary, it records that substring as a new dictionary entry; upon completion of the parse, the dictionary size $N_w(x)$ provides a raw measure of complexity. Intuitively, strings composed of a small number of repeated subpatterns yield small dictionaries and thus low complexity, whereas strings exhibiting many novel substrings produce large dictionaries and high complexity. We use the definition of LZ-complexity from (Dingle et al., 2018) (see Supplementary Note 7)

$$K_{LZ}(x) \;=\; \frac{\log_2(n)}{2}\big[N_w(x_{1\ldots n}) + N_w(x_{n\ldots 1})\big], \tag{7}$$

i.e. the average of the forward and reverse parses, which increases the number of distinct complexity values assignable to strings of a given length.

The Lempel–Ziv complexity, $K_{LZ}(x)$, satisfies a number of important asymptotic and finite-size scaling laws. For an ergodic source and in the limit $n \to \infty$, one has

$$\lim_{n \to \infty} \frac{N_w(x) \log_2 n}{n} \;=\; h(x), \tag{8}$$

where $h(x)$ is the Shannon entropy rate of the source, and consequently $K_{LZ}(x)/n \to h(x)$ for almost all long strings (Dingle et al., 2018). Complexity is bounded above by entropy, so strings of low entropy cannot exhibit high $K_{LZ}(x)$, while high-entropy strings may nonetheless have simple structure and thus low $K_{LZ}(x)$. Empirically, for short to moderate $n$, $K_{LZ}(x)$ often outperforms generic lossless compressors in approximating Kolmogorov complexity (Dingle et al., 2018). As $n$ increases, the mean and median of the normalised complexity distribution approach unity, and the relative standard deviation $\sigma/\mu$ decreases, indicating that typical complexities concentrate sharply around their mean. Strings whose complexity lies well below the mean become exponentially rare in $n$, although a small fraction of "maximally complex" strings arise anomalously via simple LZ-specific constructions. Additive and multiplicative constants in $K_{LZ}(x)$ may be absorbed into fitting parameters when modelling such simplicity-bias phenomena.

## B.2 RELATING DNF COMPLEXITY TO WEIGHT NORM

Each nonzero entry in $W^{(1)}$ corresponds bijectively to a literal in some conjunctive clause of the DNF (Proposition 2.7). Hence

$$\|W^{(1)}\|_1 \;=\; \sum_{i,j} \mathbb{1}[W^{(1)}_{ij} \neq 0] \;=\; L(\Phi_f). \tag{9}$$

Minimising this quantity gives us the complexity measure used in the main text, $K(f)$ (Proposition 2.9). However, strictly speaking, this is not the true norm of the DFCN, which must take into account the second layer: $\|\theta\|_1 := \|W^{(1)}\|_1 + \|W^{(2)}\|_1$. We define a second type of DNF-related complexity

$$K_\theta(f) = \min_{W^{(1)}, W^{(2)}} \left(\|W^{(1)}\|_1 + \|W^{(2)}\|_1\right) \tag{10}$$

Using this norm is equivalent to defining the DNF complexity as the number of literals plus the number of clauses in the minimum representation.

**Relating $K(f)$ to $K_\theta(f)$** We can lower bound the number of clauses as a function of the number of literals by creating $\lceil K(f)/n \rceil$ unique clauses with at most $n$ elements. We can upper bound this by remembering that if clauses span $k$ columns in $W^{(1)}$, we can have no more than $2^{k-1}$ clauses in the minimum representation. This means we can never have more than $2^{\lceil k \rceil - 1}$ clauses, where $k$ satisfies $K(f) = k2^{k-1}$. We can rearrange and take logs to obtain

$$K(f) + \lceil K(f)/n \rceil \leq K_\theta(f) \leq K(f) + 2^{\lceil 1 + \log_2 K(f) \rceil - 1}. \tag{11}$$

The maximum of each complexity is parity, $K(f) = \frac{n}{2}2^n$ and $K_\theta(f) = \frac{n+1}{2}2^n$.

**Problems with $K(f)$ and $K_\theta(f)$** The maximum complexity for these two measures is $O(n2^n)$: ideally, the maximum function should not have complexity greater than $2^n + O(1)$. Assuming complexity is related to compression – that is, simple functions are highly compressible – we should not "compress" a function to more bits than its string representation, which needs $2^n$ bits. One complexity measure that does satisfy this requirement is double the total number of clauses

$$K_C(f) = 2\|W^{(2)}\|_1, \tag{12}$$

which has a maximum complexity of exactly $2^n$. $\lceil K(f)/n \rceil$ provides the minimum number of clauses we can fit $K(f)$ literals in, and we can again use the fact that if the maximum clause has length $k$, we can have no more than $2^{k-1}$ clauses in the minimum representation to upper bound,

$$\lceil K(f)/n \rceil \le K_C(f) \le 2^{\lceil 1+\log_2 K(f) \rceil - 1}. \tag{13}$$

**Desirable properties** We can make some of these observations precise using known results. Blais & Tan (2015) lists the most important. The Korshunov-Kuznetsov Theorem states that a random boolean function requires $\Theta(2^n/\log n \log\log n)$ clauses, each with an expected $n - \Theta(\log n + \log\log n)$ literals. From this, we have the following scaling laws.

| Complexity | constant | $t$-entropy | $k$-parity | Random | Parity |
|---|---|---|---|---|---|
| $K(f)$ | $O(1)$ | $O(nt)$ | $k2^{k-1}$ | $\Theta\left(\frac{n2^n}{\log n \log\log n}\right)$ | $n2^{n-1}$ |
| $K_\theta(f)$ | $O(1)$ | $O((n+1)t)$ | $(k+1)2^{k-1}$ | $\Theta\left(\frac{(n+1)2^n}{\log n \log\log n}\right)$ | $(n+1)2^{n-1}$ |
| $K_C(f)$ | $O(1)$ | $O(2t)$ | $2^k$ | $\Theta\left(\frac{2^n}{\log n \log\log n}\right)$ | $2^n$ |

So from an "optimal compression" point of view, $K_C(f)$ is the best measure, $K_\theta(f)$ is the most sensible for the DFCN, and $K(f)$ is often the easiest to work with. See Figure 5 for empirical results with small $n$.

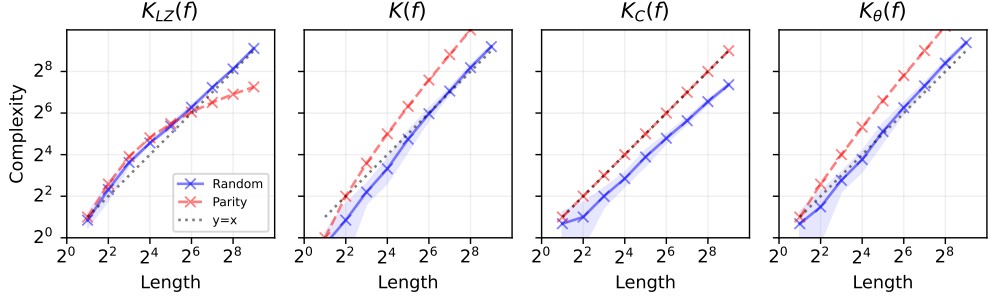

Figure 5: **Scaling of** $K_{LZ}(f), K(f), K_\theta(f), K_C(f)$ for random Boolean functions and the parity function for $n = 2$ to $n = 9$ (the length of the string representation of $f$ is therefore $2^n$). We expect random functions to be incompressible, and thus have a complexity $\approx 2^n$ for good complexity measures. LZ complexity is known to satisfy this requirement up to $O(1)$ terms (Lempel & Ziv, 1976), and for $K_C(f)$ the worst case is exactly $2^n$. For $K(f), K_\theta(f)$ however, the worst case (parity) is $\frac{n}{2}2^n$ and $\frac{n+1}{2}2^n$, respectively, and whilst the typical random functions appear to have complexities close to $2^n$ for small $n$, theoretical results in Appendix B.2 show that this would change as $n$ increases.

### B.3 Important differences between $K(f)$ and $K_{LZ}(f)$

One important class of functions where the two measures differ significantly is functions with repeating patterns. Consider the string representation of $f$.

1. Consider the function $f = $ `"1001"` $\times 2^{n-2}$. This function is 2-sparse, represented by the DNF $(\neg x_1 \wedge x_2)$ (1 clause, 2 literals). As $n$ increases, its DNF complexity remains fixed at 2. It's Lempel-Ziv complexity $K_{LZ}(f) = C + \log_2 n$ (constant term $C$ for encoding the repeating string and the $\log n$ term from the repetitions)

2. However, if we generate a function $f$ by repeating the string `"01001"` (truncating at the end) is not a $k$-sparse function. As a result, its $K(f)$ will not be constant.

See Figure 6 for empirical data showing the discrepancy between these measures.

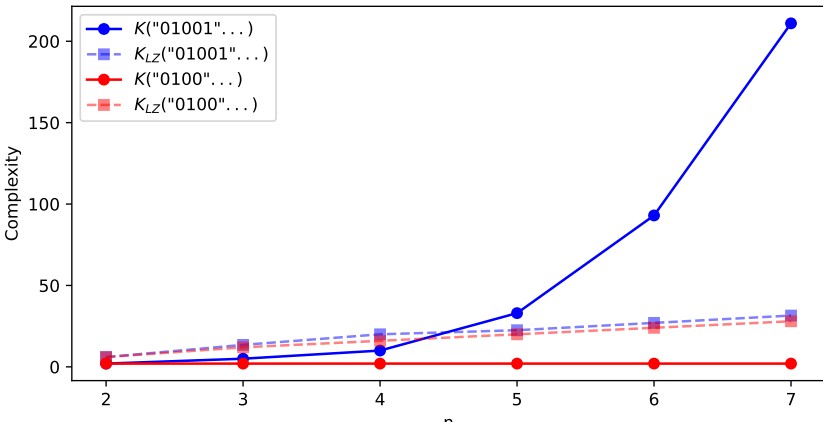

Figure 6: $K(f)$ **v.s.** $K_{LZ}(f)$ **on repeating functions**. **Dashed** lines show $K_{LZ}(f)$ and **solid** lines show $K(f)$. The **red** curves show the 2-sparse function $f = $ `"1001"` $\times 2^{n-2}$. $K(f)$ remains constant at 2, and $K_{LZ}(f)$ grows slowly with $n$. By contrast the **blue** curves show the function generated by the repeated pattern `"01001"` which yields a function that is not 2-sparse and therefore does not enjoy a small, constant $K(f)$, but does enjoy a $K_{LZ}(f)$ that grows at a similar rate to the $K_{LZ}(f)$ of the 2-sparse function.

Empirical work in (Valle-Pérez et al., 2018) shows that $K(f)$ and $K_{LZ}(f)$ are correlated. However, they have very different maximum complexities. Given that a function can be represented in its string representation with $2^n$ bits – a really good complexity measure should never go above $2^n + O(1)$. The maximum value of $K(f)$ is $\frac{n}{2}2^n$, or $O(n2^n)$. In contrast, $K_{LZ}(f)$ has a worst-case of $O(2^n)$.

Figure 5 shows how the two measures scale for random boolean functions (generated by assigning a 1 or 0 randomly for each input $x$) and parity, as a function of $n$. Random functions should have complexities close to $2^n$ (as they are not compressible beyond their string representation), which is the case for both $K_{LZ}(f)$ and $K(f)$. Parity, on the other hand, scales very differently. Future work could determine the fraction of functions with complexity greater than $2^n$.

### B.4 ZIPF'S LAW

Valle-Pérez et al. (2018) showed that the prior of neural networks is well-described by Zipf's law. For a dataset of size $2^n$, the probability of randomly initialising to a function $P(f)$ is a function of the rank of the function $R(f)$ (where the rank of the most probable function is 1, the second most is 2 and so on) that satisfies

$$P(f) = \frac{1}{2^n \log 2} \frac{1}{R(f)}, \tag{14}$$

where the first term is a normalisation term given that there are $2^{2^n}$ functions. Ridout et al. (2024) show that when $P(f)$ satisfies Zipf's law, a Bayesian learning agent on this prior will learn optimally (for a full discussion on what optimal means, see (Ridout et al., 2024)). We find it a useful reference point when considering the scaling laws in Table 2.

If Zipf's law is satisfied, the most frequent functions (in our work, the constant functions) should have $P(\text{const}) \sim 2^{-n}$. There are $2^n$ unique functions with $t = 1$ (a single True input), so these functions occupy a total space of size $\frac{1}{2^n \log 2} \sum_{i=3}^{i=2^n+2} i^{-1} \sim 2^{-n}n$, meaning the average 1-entropy function has probability $P(f_1^{(e)}) \sim n2^{-2n}$. (Note that $i = 1, 2$ correspond to the two constant functions, and we ignore the flipped entropy functions that have only a single zero, which would be of the same order but only include an extra factor of 1/2, not affecting the overall approximation). We will use these results in Appendix C.1.

## B.5   DFCNs vs FCNs

The DFCN is not a typical discrete approximation of a neural network, as the bias terms are all functions of weights, to make sure each neuron represents a clause. One consequence is that $P(f)$ is strongly width-dependent, unlike standard FCNs on Boolean data (Valle-Pérez et al., 2018). In the limit of infinite width ($\alpha_w \to \infty$), at initialisation, $P(f)$ will be entirely dominated by a constant function (Lemma D.5). This is because adding more clauses can only set more inputs to True – eventually, every input will be covered by at least one clause. See Figure 8 for an empirical demonstration.

## B.6   Why does parity generalise badly?

In Section 4.3 we discuss details about training DFCNs on $k$-parity functions. An interesting result is that for the most complex function (7-parity), the test accuracy gets worse the larger the training set. This can be understood by thinking about what a network will predict on unseen test data. Consider an example for $n = 4$ where the target function is 4-parity: `"0110100110010110"`. Let's assume in the following examples that we train to the minimum norm solution.

1. Suppose that we train on the first four bits ($m = 4$). The minimum norm solution for this is 2-parity (parity on the first two bits). Then, $f$ is just the first 4 bits repeated, 4 x `"0110"` = `"0110011001100110"`. On the remaining 12 bits, our accuracy is 33%.

2. Now we train on the first 8 bits. The minimum norm solution is now 3-parity, 2 x `"01101001"` = `"0110100101101001"`, which has 0% test accuracy.

This argument can be straightforwardly generalised to larger $n$. Choosing training examples in this way, when trying to learn parity generalisation, will get worse the larger the training set.

## B.7   Proofs from Mingard et al. (2019)

**Proof of proposition 2.7**

*Proof.* Fix an input dimension $n$. Let $f : \{0,1\}^n \to \{0,1\}$ be an arbitrary Boolean function and set

$$t = \left| \left\{ \mathbf{v} \in \{0,1\}^n \ : \ f(\mathbf{v}) = 1 \right\} \right|. \tag{15}$$

We adopt the notation for a clause $C$ as laid out in Definition 2.3.

Define the set of all DNFs by $\mathcal{F}_{\text{DNF}}$ and an equivalence relation $R_{\text{DNF}}$ to be permutations of DNF clauses such that the set of equivalence classes is $\mathcal{F}_{\text{DNF}}/R_{\text{DNF}}$. Similarly, define the set of all DFCNs of width $2^{n-1}$ by $\mathcal{F}_{\text{DFCN}}$ and an equivalence relation $\mathcal{F}_{\text{DFCN}}$ to be the permutations of rows in $W^{(1)}$ of a DFCN such that the set of equivalence classes is $\mathcal{F}_{\text{DFCN}}/R_{\text{DFCN}}$. We now show that there exists a bijective map $\mathcal{G} : \mathcal{F}_{\text{DNF}}/R_{\text{DNF}} \to \mathcal{F}_{\text{DFCN}}/R_{\text{DFCN}}$.

$\mathcal{G}$ **is injective**   We begin by assuming that $t \le 2^{n-1}$. Define a depth-two DFCN with layer sizes $\langle n, t, 1 \rangle$ and $\beta = 1$ (following Definition 2.6). For $i \in \{1, ..., t\}$, let $\mathbf{v}^{(i)}$ be the $i$-th input vector for which $f$ outputs True such that $\gamma_i = \sum_j v_j^{(i)}$ is the number of positive literals in clause $C_i$. Set

$$W_{ij}^{(1)} = \begin{cases} +1 & \text{if } v_j^{(i)} = 1, \\ -1 & \text{if } v_j^{(i)} = 0, \end{cases} \qquad b_i^{(1)} = 1 - \gamma_i, \tag{16}$$

$$W_i^{(2)} = 1, \qquad b^{(2)} = 0. \tag{17}$$

For any $\mathbf{v} \in \{0,1\}^n$,

$$z_i(\mathbf{v}) = \sum_i W_{ij}^{(1)} v_j + b_i^{(1)} \ = \ \gamma_i - d(\mathbf{v}, \mathbf{v}^{(i)}) + (1 - \gamma_i) \ = \ 1 - d(\mathbf{v}, \mathbf{v}^{(i)}), \tag{18}$$

where $d(\cdot, \cdot)$ denotes the hamming distance. $z_i(\mathbf{v}) = 1$ iff every literal in $C_i$ is satisfied and $z_i(\mathbf{v}) \le 0$ otherwise. Since $\sigma(z_i) = \text{ReLU}(z_i) = \max(z_i, 0)$,

$$\sigma\big(z_i(\mathbf{v})\big) = C_i(\mathbf{v}) = \begin{cases} 1 & \text{if } \mathbf{v}^{(i)} = \mathbf{v}, \\ 0 & \text{if } \mathbf{v}^{(i)} \ne \mathbf{v}. \end{cases} \tag{19}$$

In other words, the output of the first layer is 1 if the clause $C_i$ is satisfied and 0 not. $W^{(2)}$ then effectively acts as an OR operator, giving us

$$f_\theta(\mathbf{v}) = \mathbb{1}[W^{(2)}\,\sigma(W^{(1)}\mathbf{v} + b^{(1)}) + b^{(2)} > 0] \tag{20}$$

$$= C_1(\mathbf{v}) \vee \cdots \vee C_t(\mathbf{v}) \;=\; \Phi_f(\mathbf{v}) \;=\; f(\mathbf{v}). \tag{21}$$

If $t > 2^{n-1}$, we instead use a network with layer sizes $\langle n, 2^{n-1} - t, 1 \rangle$ and let $\mathbf{v}^{(1)}$ be the $i$-th input vector for which $f$ outputs False (which must be $\leq 2^{n-1}$). We then set $\beta = -1$, which negates $W_i^{(2)}$ and sets $b^{(2)} = 1$, giving us the following parameters,

$$W_{ij}^{(1)} = \begin{cases} +1 & \text{if } v_j^{(i)} = 1, \\ -1 & \text{if } v_j^{(i)} = 0, \end{cases} \qquad\qquad b_i^{(1)} = 1 - \gamma_i, \tag{22}$$

$$W_i^{(2)} = -1, \qquad\qquad b^{(2)} = 1. \tag{23}$$

$z_i$ still outputs 1 if the clause $C_i$ is satisfied and 0 if not, but $W^{(2)}\mathbf{z} < 0$ if any of the False clauses are satisfied. Thus, the only way to obtain a positive output inside the indicator function in Equation (20) is if all $C_i$ are not satisfied (since $b^{(2)} = 1$ brings the value above 0 in this case). This then gives us

$$f_\theta(\mathbf{v}) = \mathbb{1}[W^{(2)}\,\sigma(W^{(1)}\mathbf{v} + b^{(1)}) + b^{(2)} > 0] \tag{24}$$

$$= \neg\,[C_1(\mathbf{v}) \vee \cdots \vee C_t(\mathbf{v})] \;=\; \Phi_f(\mathbf{v}) \;=\; f(\mathbf{v}). \tag{25}$$

We can pad the width of the network to be $2^{n-1}$ by adding rows of zeros to $W^{(1)}$ and setting the rest of the weights according to Definition 2.6. We have extra degrees of freedom in the permutations of these rows and thus invoke the equivalence relation $R_{\text{DFCN}}$, proving injectivity of $\mathcal{G}$.

$\mathcal{G}$ **is surjective**  Conversely, let a parameter tensor $\theta = (W^{(1)}, b^{(1)}, W^{(2)}, b^{(2)}, \beta)$ satisfy the constraints of Table 1 with hidden width $2^{n-1}$. Keep only the indices $i$ for which $W_i^{(2)} = \beta$; there are $t \leq 2^{n-1}$ of them. For each such $i$, define the clause

$$C_i = \Big( \bigwedge_{\{j : W_{ij}^{(1)} = +1\}} x_j \Big) \bigwedge \Big( \bigwedge_{\{j : W_{ij}^{(1)} = -1\}} \neg x_j \Big). \tag{26}$$

Following the same reasoning as before, we conclude that

$$f_\theta(\mathbf{v}) = \mathbb{1}[W^{(2)}\,\sigma(W^{(1)}\mathbf{v} + b^{(1)}) + b^{(2)} > 0] = \begin{cases} C_1(\mathbf{v}) \vee \cdots \vee C_t(\mathbf{v}) & \text{if } \beta = 1, \\ \neg\,[C_1(\mathbf{v}) \vee \cdots \vee C_t(\mathbf{v})] & \text{if } \beta = -1. \end{cases} \tag{27}$$

Thus, every image has a preimage, which further holds true for the equivalence relations imposed on the DNFs and DFCNs, proving that $\mathcal{G}$ is surjectivity.

Bijectivity of $\mathcal{G}$ follows from injectivity and surjectivity. See Appendix G of (Mingard et al., 2019) for further details. $\qquad\square$

## C   APPROXIMATING $P(f)$ BY SAMPLING

We approximate $P(f)$ by Monte Carlo sampling from the uniform prior on network parameters, defined in Equation (6):

$$W_{ij}^{(1)} \sim \mathrm{U}\{-1, 0, 1\}, \quad \beta \sim \mathrm{U}[-1, 1], \quad W^{(2)} = \beta \text{ (unless the corresponding clause is zero)}, \tag{28}$$

and estimate

$$P(f) = \Pr(f_\theta = f) = \frac{|\{\theta : f_\theta = f\}|}{2 \cdot 3^{n\,2^{n-1}}}. \tag{29}$$

Because $P(f)$ counts how many choices of $\theta$ implement $f$, it is exactly equivalent to the volume of parameter-space occupied by $f$. One may also view $P(f)$ as a Bayesian prior probability assigned to $f$.

In our Monte Carlo sampling to approximate $P(f)$, we draw $10^8$ independent parameter samples. Any function $f$ with $P(f) \lesssim 10^{-8}$ is vanishingly unlikely to appear in our search. For example, when $n = 4$, one computes

$$P(\text{parity}) = (2^3)! \, 3^{-4 \cdot 2^3} \approx 2 \times 10^{-11}, \tag{30}$$

which explains why parity is never discovered (it lies three orders of magnitude below our sampling threshold). In fact, out of the $2^{2^4} = 65{,}536$ possible Boolean functions on $4$ bits, we fail to encounter $631$ of them even after $10^8$ draws.

Figure 7 plots the estimated $P(f)$ for $n = 3, 4, 5, 7$.

- **Top three rows:** $P(f)$ vs. $K(f)$, $K_\theta(f)$ and $K_C(f)$. The horizontal line at $P = 10^{-8}$ marks our effective sampling cutoff; finite-size artefacts appear for large $K(f)$, especially when $n = 7$. For $n = 3, 4$ there is a clear log-linear relationship between $P(f)$ and all complexity measures, with the difference between the maximum and minimum $P(f)$ at a fixed complexity small relative to the overall range of $P(f)$. For $n = 5, 7$, the total range of $P(f)$ is many orders of magnitude more than we can sample, but the upper bound $P(f) \sim 2^{-K(f)+O(1)}$ as observed in Valle-Pérez et al. (2018) still describes the distribution well.

- **Penultimate row:** $P(f)$ vs. $K_{LZ}(f)$, the Lempel–Ziv string complexity, as studied in Mingard et al. (2025; 2019); Valle-Pérez et al. (2018). The relation between $P(f)$ and $K_{LZ}(f)$ for $n = 3, 4$ is much weaker than for DNF complexity $K(f)$. This is what we might expect, given that DNF complexity is intuitively more appropriate in this case (as $K(f)$ is intricately connected to the architecture in a way $K_{LZ}(f)$ is not, see Appendix B.3). For $n = 5, 7$, we are unable to sample enough times to properly compare the distributions.

- **Bottom row:** $P(f)$ vs. rank $R(f)$ (where the most probable function has $R = 1$). The dashed orange line shows Zipf's law, $P(f) = (2^n \ln 2)^{-1} R(f)^{-1}$, which Ridout et al. (2024) identifies as the optimal prior for Bayesian learning.

Figure 8 shows the effect of the width of the hidden layer on the prior. We show widths $\alpha_w 2^{n-1}$ for $\alpha_w = 0.5, 1, 2, 4$, with the top two rows showing $n = 4$ and the bottom two rows showing $n = 5$. As the width increases, the probability that any input is True increases. We can use Lemma D.4 to show that the probability that any input is False scales as $(1 - (2^n - 1)/3^n)^{\alpha_w 2^{n-1}}$. This expression decreases asymptotically to 0 as $\alpha_w$ increases. The **bottom row** in Figure 8 shows that the prior is not well-described by Zipf's law for $\alpha_w > 1$, indicating $\alpha_w = 1$ gives the optimal width for learning (Ridout et al., 2024). This is an interesting coincidence which we explore in the remainder of this section.

## C.1   FINDING THE OPTIMUM WIDTH

As stated in the main text, we require a width of $\alpha_w 2^{n-1}$ with $\alpha_w \geq 1$ to guarantee full expressivity. However, Table 2 tells us that with this scaling, eventually the function space will be entirely dominated by the constant functions unless $\alpha_w \sim (3/4)^n$. Furthermore, it would be completely impractical to use a DFCN with width $2^{n-1}$ – by $n = 50$ to be fully expressive, you would need more than $10^{14}$ neurons.

So, how could we determine the optimum scaling? We assume that the presence of Zipf's law indicates an optimal prior. We argue this case in Appendix B.4, and will also make use of the derived

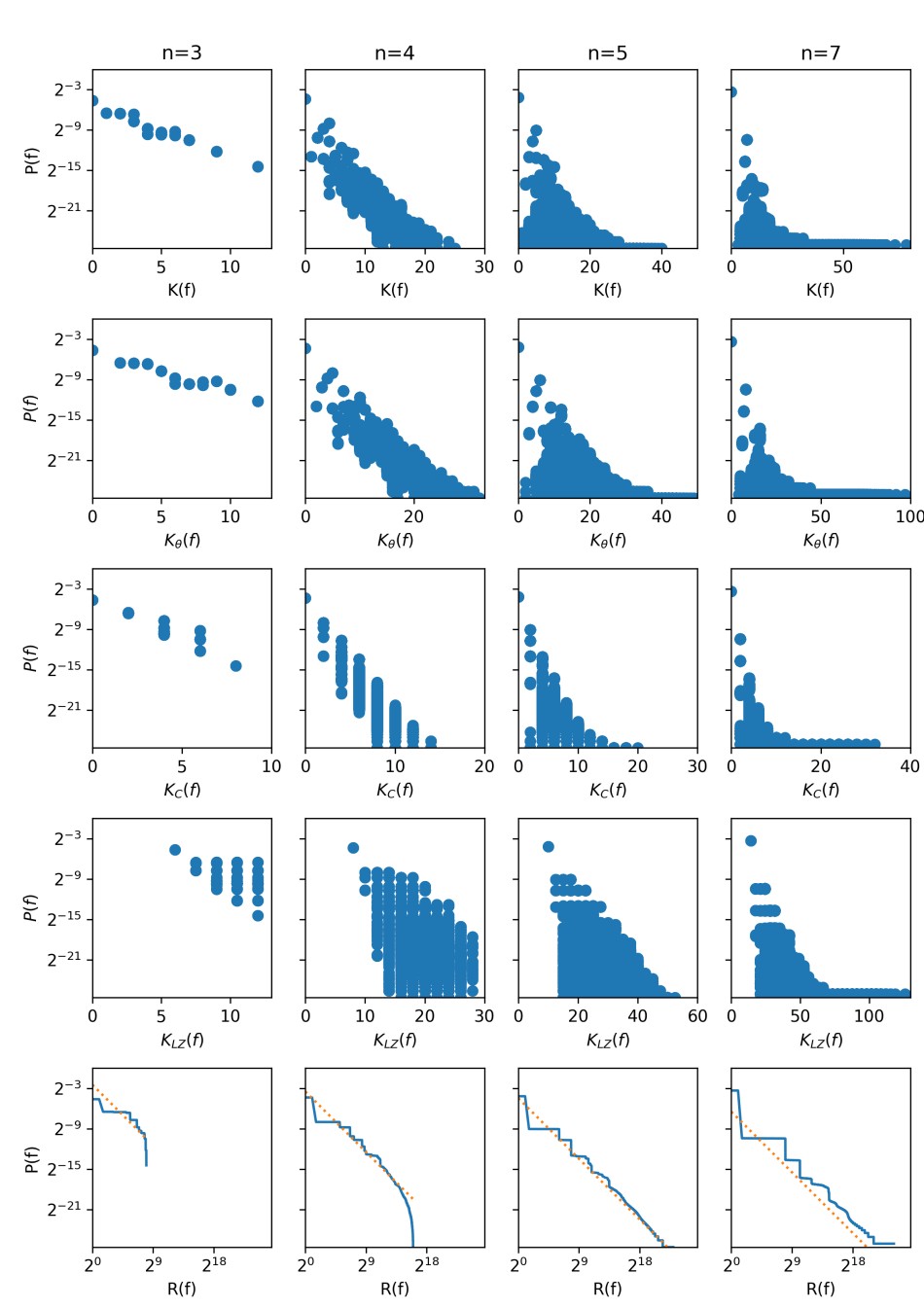

Figure 7: Approximation of the prior probability $P(f)$ by sampling $10^8$ functions from each prior for $n = 3, 4, 5,$ and $7$. **Top row:** $P(f)$ versus DNF complexity $K(f)$. Finite-size effects at $P(f) = 10^{-8}$ (sampling limit) produce artefacts at higher $K(f)$ in the $n = 7$ panel. **Second row:** $P(f)$ versus neural network norm $K_\theta(f)$. **Third row:** $P(f)$ versus clause complexity $K(f)$. **Fourth row:** $P(f)$ versus Lempel–Ziv complexity $K_{LZ}(f)$, as used in Mingard et al. (2025; 2019); Valle-Pérez et al. (2018). **Final row:** $P(f)$ versus rank $R(f)$, with dotted orange lines showing Zipf's law $P(f) = (2^n \ln 2)^{-1} R^{-1}$ (Ridout et al., 2024).

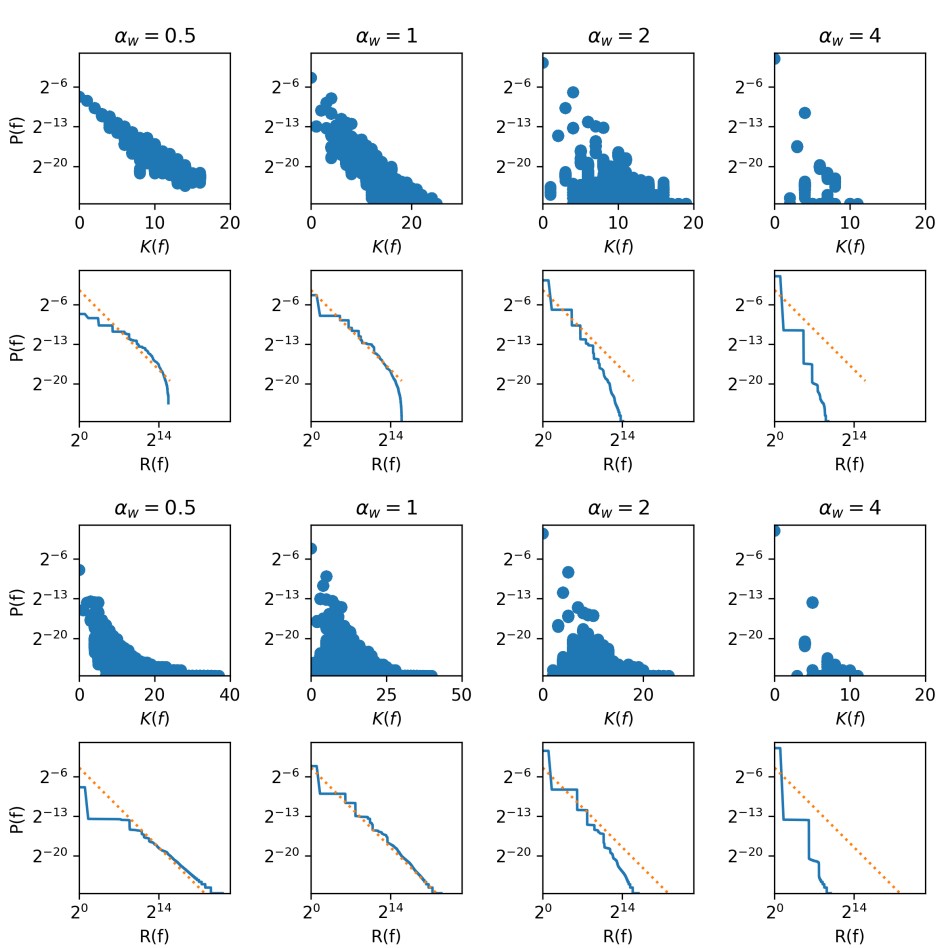

Figure 8: Coverage of the prior probability $P(f)$ for **top two rows** $n = 4$ and **bottom two rows** $n = 5$ across three network widths $w \in \{1, 2, 4\} \times 2^{n-1}$, estimated by sampling $10^8$ functions per prior. The $P(f)$ versus DNF complexity $K(f)$, illu plots illustrate how the constant function's probability mass grows with width. The plots showing $P(f)$ versus rank $R(f)$ (most probable is $R = 1$), with the dotted line marking Zipf's law $P(f) \propto R^{-1}$; larger widths exhibit marked departures from this scaling.

results $P(\text{const}) \sim 2^{-n}$ and $P(f_1^{(e)}) \sim n2^{-2n}$. We use the result from Equation (79) with $p = \frac{2^n - 1}{3^n}$,

$$P(f_1^{(e)}) \lesssim (1-p)^{\alpha_w 2^{n-1}} \tag{31}$$

$$\approx \exp(-\alpha_w 2^{n-1} p) \quad (p \ll 1) \tag{32}$$

$$\implies \quad \exp(-\alpha_w 2^{n-1} p) = n \, 2^{-n} \tag{33}$$

$$-\alpha_w 2^{n-1} p = \ln(n \, 2^{-n}) = -n \ln 2 + \ln n \tag{34}$$

$$\alpha_w 2^{n-1} = \frac{n \ln 2 - \ln n}{p} \sim n \ln 2 \left(\tfrac{3}{2}\right)^n \tag{35}$$

$$\alpha_w \sim 2n \ln 2 \left(\tfrac{3}{4}\right)^n. \tag{36}$$

We can also determine the width scaling to make the lower bound on $P(f^{(c)})$ scale as $2^{-n}$ by using the result in Lemma D.4 that the probability that an input $\mathbf{v}$ is covered with probability $(1-p)^{\alpha_w 2^{n-1}}$, where $p = \frac{2^n - 1}{3^n}$. Assuming independence and taking the Poisson approximation, we get

$$P(f^{(c)}) \approx \exp\left(-2^n e^{-\alpha_w 2^{n-1} p}\right) = 2^{-n} = \exp\left(-n \ln 2\right) \tag{37}$$

$$\implies \quad 2^n e^{-\alpha_w 2^{n-1} p} = n \ln 2 \tag{38}$$

$$e^{-\alpha_w 2^{n-1} p} = \frac{n \ln 2}{2^n} \tag{39}$$

$$-\alpha_w 2^{n-1} p = \ln\left(n \ln 2\right) - n \ln 2 \tag{40}$$

$$\alpha_w 2^{n-1} = \frac{n \ln 2 - \ln\left(n \ln 2\right)}{p} \approx (n \ln 2) \left(\tfrac{3}{2}\right)^n \tag{41}$$

$$\alpha_w \sim 2n \ln 2 \left(\tfrac{3}{4}\right)^n. \tag{42}$$

Both methods arrive at the same scaling for $\alpha_w \sim n \left(\tfrac{3}{4}\right)^n$. At small $n$, this scaling is very close to 1 – explaining why Figure 8 shows Zipf's law for $\alpha_w = 1$. To test the proposed optimal scaling more thoroughly, one would have to either make the scaling arguments above more precise or gather empirical evidence at larger $n$.

# D    RESULTS RELATING $P(f)$ TO $K(f)$

In this appendix, we relate $P(f)$ to $K(f)$ for the various classes of functions discussed in the main text (Section 3.3): Constant, $k$-parity and $t$-entropy functions. We also briefly look at $k$-sparse functions, which are not studied in the main text. The width of the DFCN is $\alpha_w 2^{n-1}$, where $\alpha_w \geq 1$ for the network to be fully expressive.

## D.1    UTILITY LEMMAS

**Lemma D.1** (Lower bound on $P(f)$). *We can lower bound $P(f)$ using*

$$P(f) \geq \frac{1}{2} P(f \mid \beta) \tag{43}$$

*as $P(\beta = 1) = P(\beta = -1) = \frac{1}{2}$*

**Lemma D.2** (Lower bound on $P(f \mid \beta)$ using the minimum representation). *Given a value for $\beta \in \{-1, 1\}$, we have the following lower bound on the conditional probability of $f$:*

$$P(f \mid \beta) \geq 3^{-nj} p! \left(\frac{p}{3^k}\right)^{\alpha_w 2^{n-1} - j}, \tag{44}$$

*where $j$ is the number of clauses, each of at most length $k$.*

*Proof.* After permuting rows, the minimum representation of a function $f$ can be encoded as follows,

$$W^{(1)} = \left[ \begin{array}{c|c} \underbrace{A}_{j \times k} & B \\ \hline C & D \end{array} \right] \Big\} \alpha_w 2^{n-1}, \tag{45}$$

$$\underbrace{\phantom{\left[ \begin{array}{c|c} A & B \\ \hline C & D \end{array} \right]}}_{n}$$

$$W^{(2)} = \left[ \underbrace{1, \ldots, 1}_{j}, \underbrace{0, \ldots, 0}_{\alpha_w 2^{n-1} - j} \right]^T \tag{46}$$

$$\beta \in \{-1, 1\} \tag{47}$$

where $A$ contains $p$ clauses each of at most length $k$, $B = 0$, $C = 0$ and $D = 0$. We can vary the clauses (rows) in $A$ only up to permutation. How much freedom do we have to vary $B, C, D$? We must set $B = 0$, otherwise $k$ would not be the maximum length of the minimal representation. If every clause in $C$ is a copy of one in $A$ and $B = 0$, we can let $D$ be anything without affecting the overall function. This gives us the following lower bound:

$$P(f \mid \beta) \geq \underbrace{j! 3^{-jk}}_{A} \times \underbrace{3^{-j(n-k)}}_{B} \times \underbrace{(j/3^k)^{\alpha_w 2^{n-1} - j}}_{C} \times \underbrace{1}_{D} \tag{48}$$

$$\geq 3^{-nj} j! \left( \frac{j}{3^k} \right)^{\alpha_w 2^{n-1} - j}. \tag{49}$$

$\square$

**Lemma D.3.** *Denote $\mathcal{N}$ the set of all possible clauses given a $\beta \in \{-1, 1\}$, with $N = |\mathcal{N}| = 3^n$. Let $M = \alpha_w 2^{n-1}$ be the number of clauses drawn i.i.d. uniformly from $\mathcal{N}$. Consider a subset $Q \subseteq \mathcal{N}$ of size $q = |Q|$ containing clauses that we must have at least one copy of, and a disjoint subset of clauses $R \subseteq \mathcal{N}$ of size $r = |R|$ that we must not have. We can then lower bound $P(f)$ as follows:*

$$P(f \mid \beta) = P = \Pr\big(\text{all } x \in Q \text{ appear at least once, and no } y \in R \text{ appears}\big), \tag{50}$$

*where*

$$P = \sum_{i=0}^{q} (-1)^i \binom{q}{i} \left( 1 - \frac{r+i}{N} \right)^{\alpha_w 2^{n-1}}, \tag{51}$$

*and the union-bound lower bound is given by*

$$P(f \mid \beta) \geq \left( \frac{N-r}{N} \right)^{\alpha_w 2^{n-1}} \left[ 1 - q\left( 1 - \frac{1}{N-r} \right)^{\alpha_w 2^{n-1}} \right]. \tag{52}$$

*In the rest of this section, we will rely on the first term in Equation (52), but as the second term is often very loose, we will often bound it below using an alternative task-specific bound.*

*Proof.* Let

$$A = \{\text{every } x \in Q \text{ appears}\}, \quad B = \{\text{no draw lies in } R\}. \tag{53}$$

Then

$$P = \Pr(A \cap B) = \Pr(A \mid B) \Pr(B). \tag{54}$$

Since each of the $M$ draws must avoid $R$,

$$\Pr(B) = \left( \frac{N-r}{N} \right)^M. \tag{55}$$

Conditioned on $B$, draws are uniform on the remaining $N - r$ symbols, and by the inclusion–exclusion principle

$$\Pr(A \mid B) = \sum_{i=0}^{q} (-1)^i \binom{q}{i} \left( \frac{(N-r)-i}{N-r} \right)^M. \tag{56}$$

Combining these and noting $\left(\frac{N-r}{N}\right)^M \left(\frac{N-r-i}{N-r}\right)^M = \left(\frac{N-r-i}{N}\right)^M$ yields the exact sum. Truncating after $i = 1$ gives us the union-bound of

$$\Pr(A \mid B) \geq 1 - q\left(1 - \frac{1}{N-r}\right)^M, \tag{57}$$

from which we obtain the union-bound on $P$ by multiplying with $\Pr(B)$.

$\square$

**Lemma D.4** (Probability of input $\mathbf{v}$ being True). *Given a fixed input $\mathbf{v} \in \{0,1\}^n$, the probability that a randomly sampled clause $C$ covers $\mathbf{v}$ (i.e., is True on $\mathbf{v}$) is*

$$P(C(\mathbf{v}) = 1) = \frac{2^n - 1}{3^n}. \tag{58}$$

*Given a DFCN of width $\alpha_w 2^{n-1}$, the probability that any particular input $\mathbf{v}$ is True is*

$$P(f(\mathbf{v}) = 1 \mid \beta = 1) = 1 - \left(1 - \frac{2^n - 1}{3^n}\right)^{\alpha_w 2^{n-1}} \tag{59}$$

*Moreover, the leading-order term for large $n$ is*

$$P(f(\mathbf{v}) = 1 \mid \beta = 1) \sim \frac{\alpha_w}{2}\left(\frac{4}{3}\right)^n. \tag{60}$$

*Proof.* A clause is True on input $x$ if for each variable in the input, the corresponding entry of a clause in DFCN representation is 1 if $x_i = 1$, is $-1$ if $x_i = 0$ or is 0 (but ignoring the all 0s case, which is considered False). This means we have $2^n - 1$ clauses that satisfy this criterion. Divide by the total number of clauses, $3^n$ for Equation (58). Since all clauses are drawn independently, the probability that all clauses give False on a random input $\mathbf{v}$ is

$$P(f(\mathbf{v}) = 0 \mid \beta = 1) = \left(1 - \frac{2^n - 1}{3^n}\right)^{\alpha_w 2^{n-1}}, \tag{61}$$

from which Equation (59) follows. $\square$

### D.2 FUNCTION CLASS: CONSTANT

Denote the class of constant functions as $f^{(c)}$, which includes all functions where the output either always gives True or always gives False, regardless of the input.

**Lemma D.5** (Lower bound for $P(f^{(c)})$). *We can bound either constant function, $P(f^{(c)})$ with*

$$P(f^{(c)}) \geq \frac{1}{2} \sum_{k=0}^{2^n} (-1)^k \binom{2^n}{k} \left(1 - kp\right)^{\alpha_w 2^{n-1}}, \tag{62}$$

*where $p = \left(\frac{2^n - 1}{3^n}\right)$. When truncating after $k = 1$, we obtain the lower bound*

$$P(f^{(c)}) \geq \frac{1}{2}\left(1 - 2^n(1-p)^{\alpha_w 2^{n-1}}\right) \tag{63}$$

*Substituting $p = \frac{2^n - 1}{3^n} \sim \left(\frac{2}{3}\right)^n$, we get*

$$P(f^{(c)}) \gtrsim \frac{1}{2}\left(1 - \exp\left(n \ln 2 - \frac{\alpha_w}{2}(4/3)^n\right)\right) \quad (n \to \infty). \tag{64}$$

*Proof.* Let $n \in \mathbb{N}$, $M = \alpha_w 2^{n-1}$, and $p = \frac{2^n - 1}{3^n}$ (Lemma D.4). We fix $\beta = 1$ and aim to bound the function that returns True for all inputs. We label the $2^n$ Boolean inputs by $\mathbf{v} \in \{0,1\}^n$, and for each $\mathbf{v}$ let

$$A_{\mathbf{v}} = \{\text{no clause covers } \mathbf{v}\}, \qquad \Pr(A_{\mathbf{v}}) = (1-p)^M. \tag{65}$$

Then by the principle of inclusion-exclusion, the probability that every input is covered by at least one clause (i.e. no $A_{\mathbf{v}}$ occurs) is exactly

$$P(f^{(c)} \mid \beta = 1) = \Pr\Big(\bigcap_{\mathbf{v} \in S} A_{\mathbf{v}}^c\Big) = 1 - \Pr\Big(\bigcup_{\mathbf{v} \in S} A_{\mathbf{v}}\Big) = \sum_{k=0}^{2^n} (-1)^k \sum_{\substack{S \subseteq \{0,1\}^n \\ |S| = k}} \Pr\Big(\bigcap_{\mathbf{v} \in S} A_{\mathbf{v}}\Big). \quad (66)$$

Moreover, for any fixed $S$ of size $k$, one has

$$\Pr\Big(\bigcap_{\mathbf{v} \in S} A_{\mathbf{v}}\Big) = \big(1 - p_S\big)^M, \quad (67)$$

where

$$p_S = \Pr\big(\text{a single random clause covers at least one } \mathbf{v} \in S\big)$$

Furthermore, by the union bound on the single-clause covering probabilities,

$$p_S \leq \sum_{x \in S} p = kp, \quad (68)$$

so that

$$\Pr\Big(\bigcap_{\mathbf{v} \in S} A_{\mathbf{v}}\Big) = \big(1 - p_S\big)^M \geq (1 - kp)^M. \quad (69)$$

Substituting into the inclusion–exclusion sum gives the valid lower bound

$$P(f^{(c)} \mid \beta = 1) \geq \sum_{k=0}^{2^n} (-1)^k \binom{2^n}{k} (1 - kp)^M. \quad (70)$$

Truncating after $k = 1$ gives $P(f^{(c)} \mid \beta = 1) \geq 1 - 2^n(1-p)^M$, and approximating $p$ to be small, so that $(1-p)^M \approx e^{-Mp}$, yields the final estimate for $P$,

$$P(f^{(c)} \mid \beta = 1) \geq 1 - 2^n(1-p)^{\alpha_w 2^{n-1}} \approx 1 - 2^n e^{-\alpha_w 2^{n-1} p}. \quad (71)$$

Using Lemma D.1, we thus have $P(f^{(c)}) \geq \frac{1}{2} P(f^{(c)} \mid \beta = 1)$.

$\square$

### D.3 FUNCTION CLASS: ENTROPY

Consider the class of functions with $t$ 1s and $2^n - t$ 0s. We call this function $t$-entropy and denote it with $f_t^{(e)}$. If $t$ is small, the function is simple (consistent with the intuition that low-entropy functions are simple). However, the converse does not hold: some high-entropy functions require a very small number of clauses (e.g. 1-parity needs just one clause: $x_1$).

**Lemma D.6.** *Given a boolean function on $n$ variables with $t$ 1s and $2^n - t$ 0s, we denote $R$ the set of forbidden clauses with $r = |R|$ and $N = 3^n$ the total number of possible clauses (Lemma D.3). The fraction of clauses that would flip the function value if appended to its DNF is bounded below by:*

$$\frac{r}{N} \geq (2/3)^{n - \lfloor \log_2(2^n - t) \rfloor} \quad (72)$$

*Proof.* Let $f : \{0,1\}^n \to \{0,1\}$ have exactly $\tilde{t} = 2^n - t$ inputs $\mathbf{v}^{(i)} \in \{\mathbf{v}^{(1)}, ..., \mathbf{v}^{(\tilde{t})}\}$ for which $f(\mathbf{v}^{(i)}) = 0$, with $1 \leq \tilde{t} \leq 2^n - 1$. By filling the largest possible $d$-dimensional subspace of the $n$-dimensional hypercube, given by $d \leq \lfloor \log_2 \tilde{t} \rfloor$, this corresponds to finding the maximal correlation between these inputs, which minimises the set of not allowed clauses $R$.

Given a maximally filled $d$-dimensional subspace, the remaining $n - d$ bits for these points must be the same. Since the subspace is fully filled, all possible combinations of inputs in this subspace are exhausted, meaning that unless all entries in the DFCN representation of a clause are zero (which always gives False), one of these subspace inputs must give True. Thus, we need the probability that

all remaining $n - d$ entries in a DFCN clause either match the corresponding remaining input bit (1 if $x_i = 1$, $-1$ if $x_i = 0$) or are 0, giving

$$P = \left(\frac{2}{3}\right)^{n-d} - 1. \tag{73}$$

(The $-1$ comes from the all zeros clause.) Substituting the bound on $d$ and taking $n$ to be large gives us

$$P = \frac{r}{N} \geq \left(\frac{2}{3}\right)^{n - \lfloor \log_2 \tilde{t} \rfloor}. \tag{74}$$

$\square$

**Lemma D.7** (Upper bound for $t$-entropy). *We can upper bound $P(f_t^{(e)})$ with the following*

$$P(f_t^{(e)} \mid \beta) \lesssim \begin{cases} \exp\left(-\alpha_w 2^{n-1}(2/3)^{n-\lfloor \log_2(2^n - t) \rfloor}\right) & \text{if } \beta = 1, \\ \exp\left(-\alpha_w 2^{n-1}(2/3)^{n-\lfloor \log_2 t \rfloor}\right) & \text{if } \beta = -1. \end{cases} \tag{75}$$

*Proof.* Let $f_t^{(e)} : \{0,1\}^n \to \{0,1\}$ have exactly $t$ 1s, with $1 \leq t \leq 2^n - 1$. For the case of $t \leq 2^{n-1}$, we take $\beta = 1$ (Appendix B.7), giving us $\tilde{t} = 2^n - t$ inputs $\mathbf{v}^{(i)} \in \{\mathbf{v}^{(1)}, ..., \mathbf{v}^{(\tilde{t})}\}$ for which $f_t^{(e)}(\mathbf{v}^{(i)}) = 0$. (For $t > 2^{n-1}$ we take $\beta = -1$ and simply replace $\tilde{t}$ with $t$.) Given the set $R$, with $r = |R|$, of all forbidden clauses which would flip a 0 output to a 1, and $N = 3^n$ total clause options, a valid network must sample all $M$ clauses outside of $R$, giving us an upper bound,

$$P\left(f_t^{(e)} \mid \beta = 1\right) \leq \left(1 - \frac{r}{N}\right)^M \lesssim \exp\left(-Mr/N\right). \tag{76}$$

Using the result in Lemma D.6, we have $r/N \geq (2/3)^{n - \lfloor \log_2 \tilde{t} \rfloor}$. Substituting $M = \alpha_w 2^{n-1}$ into equation 76 then gives

$$P\left(f_t^{(e)} \mid \beta = 1\right) \leq \left(1 - (2/3)^{n - \lfloor \log_2(2^n - t) \rfloor}\right)^{\alpha_w 2^{n-1}} \approx \exp\left(-\alpha_w 2^{n-1}(2/3)^{n - \lfloor \log_2(2^n - t) \rfloor}\right). \tag{77}$$

$\square$

**Lemma D.8** (Bounds on $t$-entropy with $t = 1$). *For a function with a single True output $\mathbf{v}$ and all else False,*

$$P(f_1^{(e)} \mid \beta = -1) \leq \left(1 - \frac{2^n - 1}{3^n}\right)^{\alpha_w 2^{n-1}} \tag{78}$$

*We also have a lower bound*

$$P(f_1^{(e)} \mid \beta = -1) \geq 3^{-n^2}\left(1 - \frac{2^n - 1}{3^n}\right)^{\alpha_w 2^{n-1} - n}. \tag{79}$$

*The leading order behaviour of this (for constant $\alpha_w$) is*

$$P(f_1^{(e)} \mid \beta = -1) \gtrsim \exp\left(-\frac{\alpha_w}{2}\left(\frac{4}{3}\right)^n\right) \tag{80}$$

*Proof.* We set $\beta = -1$, so that we are now solving for a function where only one input $\mathbf{v}^{(1)} \in \{0,1\}^n$ has an output of 0. To prove the upper bound, we simply require no clause to be True on input $\mathbf{v}^{(1)}$.

To prove the lower bound, we can use the fact that the minimal representation of this function is given by a weight $W^{(1)}$ with only non-zero entries in the main diagonal,

$$W_{ii}^{(1)} = \begin{cases} +1 & \text{if } v_i^{(1)} = 0, \\ -1 & \text{if } v_i^{(1)} = 1. \end{cases} \tag{81}$$

Note this is the opposite of the typical construction where we assign $+1$ if the input bit is 1 and $-1$ if the input bit is 0. Provided none of the $2^n - 1$ clauses that make $f_1^{(e)}(\mathbf{v})$ output True are drawn (Lemma D.4), we can lower bound $P(f_1^{(e)})$ by setting the first $n$ rows of $W^{(1)}$ as described above (which would happen with probability $3^{-n^2}$) and require the rest of the rows to exclude any of the $2^n - 1$ forbidden clauses. $\square$

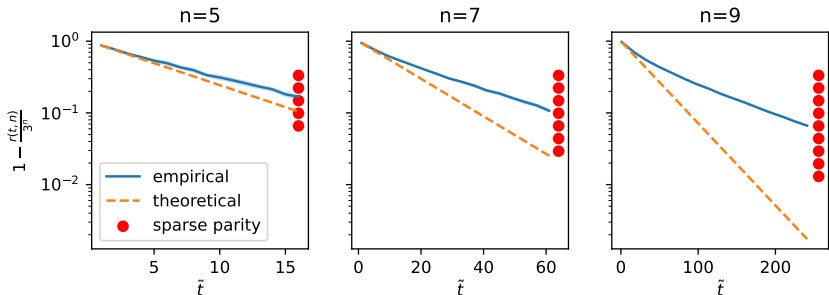

Figure 9: Fraction of accepted clauses $1 - \frac{r(n,t)}{3^n}$ versus the number of zeros, $\tilde{t}$, in an $n$-variable Boolean function. Error bars show 1 standard deviation. The theoretical line uses Equation (82), and assumes independence. This is only a good assumption for low $\tilde{t}$. As expected, when $\tilde{t}$ is small (the function is almost entirely True), almost all clauses are accepted without changing the function. The red dots indicate all $k$-parity functions for $1 \leq k \leq n$.

### D.3.1 $P(f)$ FOR A RANDOM FUNCTION WITH FIXED $t$

In Appendix D.3, we upper bounded $P(f_t^{(e)})$ by computing the minimum number of forbidden clauses at every entropy class. We know there can be a huge range in $P(f_t^{(e)})$, the best example being $t = 2^{n-1}$. Both 1-parity and $n$-parity have $2^{n-1}$ 1s, yet $P(f_n^{(p)}) \sim (3/2)^{-\alpha_w n 2^{n-1}}$ and $P(f_1^{(p)}) \sim (3/2)^{-\alpha_w 2^{n-1}}$. The bounds must satisfy $P(f_1^{(p)})$ and are therefore far too loose for $f_n^{(p)}$.

We did not, however, determine how $P(f_t^{(e)})$ should scale for the typical function with $t$ 1s (i.e. a uniformly sampled function from the set of functions with exactly t 1s). Consider a function $f$, and define the number of False outputs $\tilde{t} = 2^n - t$. The probability of drawing a clause $C$ which is True on an input $\mathbf{v}$ for which $f(\mathbf{v})$ outputs False is $p = (2^n - 1)/3^n$ (Lemma D.4). Then with $R$ as the set of all forbidden clauses and $r = |R|$, the probability of drawing a forbidden clauses $C \in R$ is

$$\Pr(C \in R) = \frac{r(n,t)}{3^n} = 1 - (1-p)^{2^n - t}, \tag{82}$$

assuming independence. Figure 9 shows that this assumption is only valid for $\tilde{t} = O(n)$. As $\tilde{t}$ increases, Equation (82) overestimates $P(C \in R)$. The empirical data was generated by uniformly sampling 20 functions at fixed $t$ and calculating $r$ by exhaustive enumeration for those functions. This gives us an idea of $r(n,t)$ for the "typical" function with $t$ 1s. We also plotted the $k$-parity functions for $1 \leq k \leq n$ (which all have $t = 2^{n-1}$).

### D.4 FUNCTION CLASS: PARITY

Let $f_k^{(p)} : \{0,1\}^n \to \{0,1\}$ be the $k$-parity function on the first $k$ bits:

$$f_k^{(p)}(\mathbf{v}) = \sum_{i=1}^{k} v_i \pmod 2, \qquad 1 \leq k \leq n. \tag{83}$$

As one checks directly, any representation of $f_k^{(p)}$ in our random network model must use exactly $j = 2^{k-1}$, distinct clauses of length $k$, and no shorter set of clauses can realise parity. With $M = \alpha_w 2^{n-1}$ and $N = 3^n$ we can construct lower and upper bounds with the following two propositions.

**Proposition D.9** (Lower bound for $k$-parity). *Using Lemma D.2, with the minimum representation size $j \times k$, set $j = 2^{k-1}$. Then,*

$$P(f_k^{(p)}) \geq 3^{-n2^{k-1}} (2^{k-1})! \left( \frac{2^{k-1}}{3^k} \right)^{\alpha_w 2^{n-1} - 2^{k-1}}. \tag{84}$$

*Applying Stirling's inequality $j! \geq \sqrt{2\pi j}(j/e)^j$, in the large $n$ limit we have*

$$P(f_k^{(p)}) \geq 3^{-n2^{k-1}}\sqrt{2\pi 2^{k-1}}\left(\frac{2^{k-1}}{e}\right)^{2^{k-1}}\left(\frac{2^{k-1}}{3^k}\right)^{\alpha_w 2^{n-1}-2^{k-1}} \tag{85}$$

$$\geq \exp\left(-\alpha_w 2^{n-1}(k\ln(3/2)+\ln 2)+O(n2^{k-1})\right). \tag{86}$$

**Proposition D.10** (Upper bound for $k$-parity)**.** *Exactly $2^{k-1}$ of the $3^k$ clauses of length $\leq k$ implement $k$-parity, so at each of the $\alpha_w 2^{n-1}$ draws the chance of choosing an admissible clause is at most $\frac{2^{k-1}}{3^k}$. Therefore*

$$P(f_k^{(p)}) \leq \left(\frac{2^{k-1}}{3^k}\right)^{\alpha_w 2^{n-1}}. \tag{87}$$

*At large $n$,*

$$P(f_k^{(p)}) \leq \exp\left(-\alpha_w 2^{n-1}(k\ln(3/2)+\ln 2)\right) \tag{88}$$

The bounds above match up to constants, so

$$P\big(f_k^{(p)}\big) = e^{-\Theta(\alpha_w k2^{n-1})}. \tag{89}$$

### D.5   FUNCTION CLASS: SPARSE

A Boolean function $f\colon \{0,1\}^n \to \{0,1\}$ is called $k$-sparse if it depends on exactly $k$ of its $n$ input bits (say $x_1, \ldots, x_k$) and is independent of the remaining $n-k$ bits. Equivalently, for every fixed $(x_1, \ldots, x_k)$, flipping any of the last $n-k$ coordinates does not change $f$. In the ordered listing of the truth table (Figure 2), $f$ then repeats its $2^k$-bit pattern $2^{n-k}$ times.

It is hard to come up with bounds for $k$-sparse functions that are meaningful, as the complexity range for a given $k$ can be very large. The most complex $k$-sparse in each class is $k$-parity. At the other end of the spectrum, there are functions where the minimum representation (Equation (45)) has $A = I_k$ (the identity matrix with dimension $k$). We could lower bound this type of function by requiring $I_k$ to exist (with probability $3^{-nk}$), and that the first $k$ elements of the rest of the clauses must not contain exclusively $-1$ and $0$ (but we permit all 0s). The probability that this happens is $\left(\frac{2}{3}\right)^k - 3^{-n}$. We can multiply this by the total number of clause options, $N = 3^n$, to give us $r = 3^n\left(\frac{2}{3}\right)^k - 1$ and then use Lemma D.3 to get,

$$3^{-nk}\left(1-\left(\tfrac{2}{3}\right)^k+3^{-n}\right)^{\alpha_w 2^{n-1}-k} \leq P(f_k^{(s)}) \leq \left(1-\left(\tfrac{2}{3}\right)^k+3^{-n}\right)^{\alpha_w 2^{n-1}}, \tag{90}$$

which to leading order scales as $\exp\left(-\alpha_w 2^{n-1}\left(\frac{2}{3}\right)^k\right)$, decaying slower than $k$-parity for large $k$. (The upper bound comes from rejecting all forbidden clauses.)

## E   TRAINED NETWORKS

### E.1   GENERATING DATASETS

Each function is a Boolean map $f : \{0,1\}^n \to \{0,1\}$, stored as an $n$-dimensional binary input vector and a scalar output. To ensure reproducibility, we set a fixed random seed at the start of generation. Given a training set size $m$, we randomly shuffle all $2^n$ possible inputs and take the first $m$ as training examples; the remaining $2^n - m$ points form the test set.

We train DFCNs on the following three functions

1. $k$-**parity:** Choose a random subset $S \subseteq \{1, \ldots, n\}$ of size $k$, and define: $f(x) = \bigoplus_{i\in S} x_i$.

2. $t$-**entropy**: Select $t$ input points uniformly at random from the $2^n$ possibilities and assign $f(x) = 1$ on those points (all others map to 0), yielding functions of fixed Hamming weight $t$.

3. $k$-**sparse:** Generate a random binary string $s \in \{0,1\}^{2^k}$, then tile it $2^{n-k}$ times to form the full function string of length $2^n$.

Note that we do not study $k$-sparse functions in the main text. These are functions generated by repeated patterns of length $2^k$.

In our experiments, we fix $n = 7$. The parameter grids are:

- $k$-parity: $k \in \{1, 2, \ldots, 7\}$.
- $t$-entropy: $t \in \{0, 4, 8, 16, 32, 35, 64\}$.
- $k$-sparse tile lengths $l \in \{2, 4, 5, 8, 13, 16, 32\}$.

Note that the repeating patterns with $l = 5, 13$ do not generate $k$-sparse functions (unlike the other lengths $l$ given above). We include them as they have low LZ complexity but not low DNF complexity (see Appendix B.3), and this example is useful in demonstrating why $K(f)$ is a better measure of complexity than $K_{LZ}(f)$.

### E.2 METROPOLIS-HASTINGS ALGORITHM

While we could in theory use an SGD-like algorithm (Algorithm 3), which aims to find the direction of steepest descent and move there, it is not Bayesian, and enumerating the entire neighbourhood rapidly increases exponentially in computational complexity as $n$ increases. In this section, we define a Metropolis-Hastings algorithm that is Bayesian, and does not suffer from these scaling problems.

Let $\theta = (W^{(1)}, W^{(2)})$ denote the parameter vector of weights in a DFCN, with

$$W^{(1)} \in \{-1, 0, 1\}^{n \times \alpha_w 2^{n-1}}, \quad W^{(2)} \in \{0, 1\}^{\alpha_w 2^{n-1}}.$$

We write $f(\theta; S)$ for the network's predictions on a dataset $S$, and $L\big(f(\theta; S)\big)$ for its empirical loss (e.g. classification error) on $S$. We also use the $\ell_1$-norm regulariser

$$\|\theta\|_1 = \|W^{(1)}\|_1 + \|W^{(2)}\|_1.$$

**Target (posterior) density.** Given inverse-temperature $\kappa > 0$ and weight-decay factor $\lambda \geq 0$, we seek to sample from

$$\pi(\theta) \propto \exp\Big[-\kappa L\big(f(\theta; S)\big) - \lambda \|\theta\|_1\Big].$$

**Proposal distribution.** For any current state $\theta$, its 1-hop neighbourhood is

$$\mathcal{N}(\theta) = \{\theta' : d(\theta, \theta') = 1\},$$

where $d(\theta, \theta')$ is the Hamming distance between discrete parameters. We use the uniform proposal

$$g(\theta \to \theta') = \begin{cases} \dfrac{1}{|\mathcal{N}(\theta)|}, & \text{if } \theta' \in \mathcal{N}(\theta), \\ 0, & \text{otherwise.} \end{cases}$$

Algorithm 1 writes down the process explicitly.

This algorithm trains well with $\kappa = 1000$ and $\lambda = 0$ – increasing $\lambda$ to 0.1 had limited effect except to destabilise early training. Figure 10 shows the outcomes of training a DFCN with Algorithm 1 on different function classes. For entropy and repeated functions, we see that adding weight decay greatly improves performance, especially when the size of the training set $m$ is smaller. As discussed in Section 4.3, weight decay does not provide any significant advantages when trying to learn highly complex functions such as 7-parity.

### E.3 MIN NORM ORACLE ALGORITHM

We also define an Oracle algorithm, which computes the minimal complexity DNF compatible with the training set. This is obtained by exhaustive search and is only possible for small enough $n$. Since this always returns the minimum norm DFCN solution, this acts as a good baseline to compare other algorithms to, as we can see how close our trained function is to having the optimal minimal complexity.

---

**Algorithm 1** Metropolis–Hastings optimisation for DFCNs

---

1: **Initialise:**
2: $W^{(1)} \sim \{-1, 0, 1\}^{n \times \alpha_w 2^{n-1}}$, $\beta \sim \{-1, 1\}$, $W^{(2)} \sim \{0, \beta\}^{\alpha_w 2^{n-1}}$
3: $b^{(1)} \leftarrow b^{(1)}(W^{(1)})$, $b^{(2)} \leftarrow b^{(2)}(\beta)$ $\qquad \triangleright$ Biases are functions of weights, see Definition 2.6
4: **Input:** Training set $S$, batch size $b = |S|$, iterations $N$ $\qquad \triangleright$ full batch if Bayesian
5: **Hyperparams:** $\kappa > 0$, $\lambda \geq 0$
6: **Initialise:** $\theta^{(0)}$ uniformly in parameter space
7: **for** $t = 1, \ldots, N$ **do**
8: $\qquad$ Sample minibatch $S_t \subset S$, $|S_t| = b$
9: $\qquad$ Propose $\theta' \sim g(\theta^{(t-1)} \rightarrow \cdot)$
10: $\qquad$ Compute losses $L_{\text{old}} = L\big(f(\theta^{(t-1)}; S_t)\big)$, $L_{\text{new}} = L\big(f(\theta'; S_t)\big)$
11: $\qquad$ Compute acceptance probability

$$\alpha = \min\Big\{1, \exp\big[\kappa\,(L_{\text{old}} - L_{\text{new}}) + \lambda\,(\|\theta^{(t-1)}\|_1 - \|\theta'\|_1)\big]\Big\}$$

12: $\qquad$ Draw $u \sim \text{Uniform}(0, 1)$
13: $\qquad$ **if** $u < \alpha$ **then**
14: $\qquad\qquad$ $\theta^{(t)} \leftarrow \theta'$
15: $\qquad$ **else**
16: $\qquad\qquad$ $\theta^{(t)} \leftarrow \theta^{(t-1)}$
17: $\qquad$ **end if**
18: **end for**

---

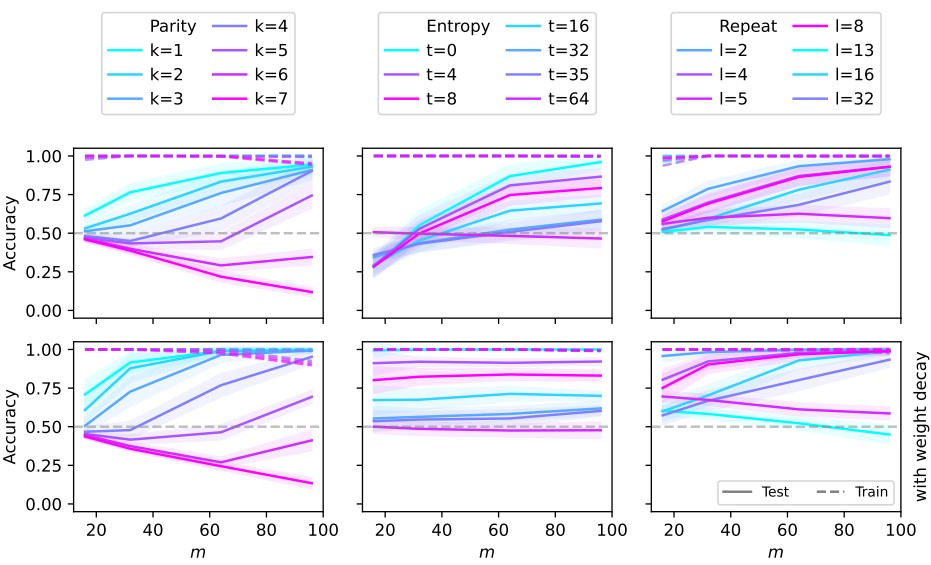

Figure 10: **MCMC algorithm** (Algorithm 1 with $\kappa = 1000$) trained on different targets from the $n = 7$ dataset. Each column shows a different function class – parity, entropy and repeat. See Appendix E.1 for full experimental details and a description of each function type. As with the SGD-like algorithm shown in Figure 12, weight decay (bottom row, $\lambda = 0.01$) outperforms no weight decay (top row $\lambda = 0$), especially for small training set size $m$.

---

**Algorithm 2** Min norm Oracle

---

1: **Initialise:**
2: True inputs $P \leftarrow \{\ \}$                            ▷ Inputs for which $f(\mathbf{v}) = 1$
3: DNF                                               ▷ Minimum DNF
4: **Input:** Training data $S$, test data $T$
5: **for** $\mathbf{s} \in S$ **do**
6:     **if** $f(\mathbf{s}) = 1$ **then**
7:         $P \leftarrow P \cup \{\mathbf{s}\}$
8:     **end if**
9: **end for**
10: DNF $\leftarrow$ SOPform(variables=$[x_1, \ldots, x_n]$, minterms=$P$, dontcares=$T$)

---

The training algorithm is shown in Algorithm 2. The SOPform function comes from the sympy logic module (Meurer et al., 2017) and finds the minimal DNF expression for a given set of inputs that output True. It takes the following arguments:

- **variables:** A list of symbols denoting the literals in the DNF.

- **minterms:** All inputs for which the output of the expression should give True.

- **dontcares:** All inputs for which we don't care about the output (i.e. the test data).

See Figure 11 for data.

### E.4 SGD-LIKE ALGORITHM

Despite not being Bayesian, we also trained with an SGD-like algorithm, which worked well for small $n$ in which the algorithm is tractable. In Algorithm 3, we begin by randomly initializing the first-layer weights $W^{(1)} \in \{-1, 0, 1\}^{n \times \alpha_w 2^{n-1}}$ and second-layer weights $W^{(2)} \in \{0, 1\}^{\alpha_w 2^{n-1}}$, and computing the dependent biases via $b^{(1)} = b^{(1)}(W^{(1)})$ and $b^{(2)} = b^{(2)}(\beta)$ (see Definition 2.6). Over $N$ iterations, we draw a minibatch $S_t$ of size $b$ from the training set, evaluate the current network accuracy on $S_t$, and enumerate all one-hop neighbours of $(W^{(1)}, W^{(2)})$. For each neighbour, we recompute its biases and measure its batch accuracy, then collect the subset $\mathcal{N}_{\text{best}}$ of weights achieving the highest performance. With probability $p$ we choose the neighbour from $\mathcal{N}_{\text{best}}$, which minimises the $\ell_1$ norm $\|W^{(1)}\|_1 + \|W^{(2)}\|_1$ (thus encouraging sparsity), and otherwise select uniformly at random from $\mathcal{N}_{\text{best}}$. The chosen weights replace $(W^{(1)}, W^{(2)})$, their biases are updated, and the procedure repeats. Upon completion, the algorithm returns a two-layer DFCN that is locally optimised for the training data.

Figure 12 shows this algorithm trained on a DFCN with $n = 7$ over a wide range of functions. We see that the performance is very similar to Algorithm 1. However, since this SGD-like algorithm does not scale well, we would instead opt to use the MCMC algorithm for large $n$.

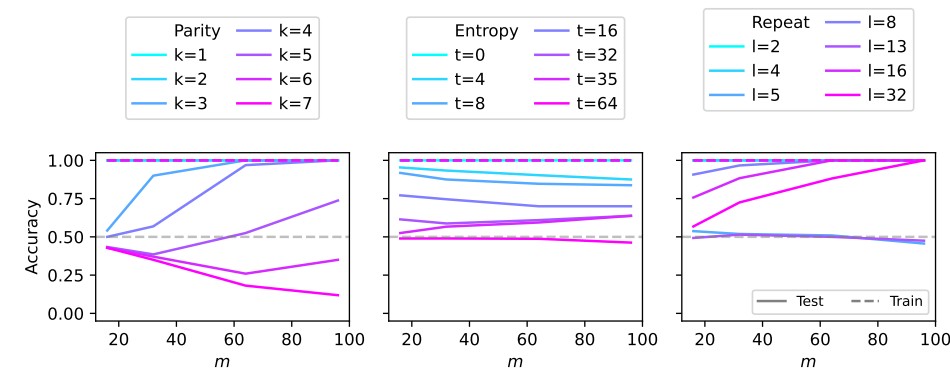

Figure 11: **The oracle** trained on different targets from the $n = 7$ dataset. Each column shows a different function class – parity, entropy and repeat. See Appendix E.1 for full experimental details and a description of each function type.

---

**Algorithm 3** SGD-like optimisation for DFCNs

---

1: **Initialise:**
2: $W^{(1)} \sim \{-1, 0, 1\}^{n \times \alpha_w 2^{n-1}}, \beta \sim \{-1, 1\}, W^{(2)} \sim \{0, \beta\}^{\alpha_w 2^{n-1}}$
3: $b^{(1)} \leftarrow b^{(1)}(W^{(1)}), b^{(2)} \leftarrow b^{(2)}(\beta)$       $\triangleright$ Biases are functions of weights, see Definition 2.6
4: **Input:** Training data $S$, test data $T$, batch size $b$, probability $p$, number of iterations $N$
5: **for** $t = 1$ to $N$ **do**
6:      Sample a batch $S_t \subset S$ of size $b$
7:      Compute current accuracy $a_{\mathrm{curr}}$ on $S_t$
8:      Generate 1-hop neighbours of $W^{(1)}$ and $W^{(2)}$
9:      **for** each neighbour $(W^{(1)\prime}, W^{(2)\prime})$ **do**
10:         **for** $\beta \in \{-1, 1\}$ **do**       $\triangleright$ We leave $\beta$ fixed at 1 in our experiments
11:            $b^{(1)\prime} \leftarrow b^{(1)}(W^{(1)\prime}), b^{(2)\prime} \leftarrow b^{(2)}(\beta)$
12:            Compute accuracy $a'$ of $(W^{(1)\prime}, b^{(1)\prime}, W^{(2)\prime}, b^{(2)\prime})$ on $S_t$
13:         **end for**
14:      **end for**
15:      Identify neighbour set $\mathcal{N}_{\mathrm{best}}$ with best accuracy
16:      **if** $r \sim U[0, 1] < p$ **then**
17:         Select $(W^{(1)\star}, W^{(2)\star})$ from $\mathcal{N}_{\mathrm{best}}$ with lowest $\|W^{(1)\star}\|_1 + \|W^{(2)\star}\|_1$ $\triangleright$ "Weight decay"
18:      **else**
19:         Select $(W^{(1)\star}, W^{(2)\star})$ uniformly at random from $\mathcal{N}_{\mathrm{best}}$
20:      **end if**
21:      Update: $W^{(1)} \leftarrow W^{(1)\star}, W^{(2)} \leftarrow W^{(2)\star}$
22:      Update: $b^{(1)} \leftarrow b^{(1)}(W^{(1)}), b^{(2)} \leftarrow b^{(2)}(\beta)$
23: **end for**

---

### E.5 RELATING ALGORITHM 3 (STEEPEST DESCENT) TO ALGORITHM 1 (METROPOLIS-HASTINGS)

Algorithm 1 and Algorithm 3 differ in two key ways.

**Probabilistic vs. deterministic greedy acceptance.** In the MCMC algorithm, we sample with probability $\alpha$, which is not the case for the steepest descent algorithm where we always evaluate all 1-hop neighbours in each batch and identify the subset $\mathcal{N}_{\mathrm{best}}$ achieving minimal loss (i.e. highest accuracy), from which the samples are drawn for the next update. We can achieve the steepest descent behaviour in the MCMC algorithm by setting $\kappa \to \infty$ (so that moves reducing the loss are always accepted), but the minimum norm selection then becomes difficult to replicate.

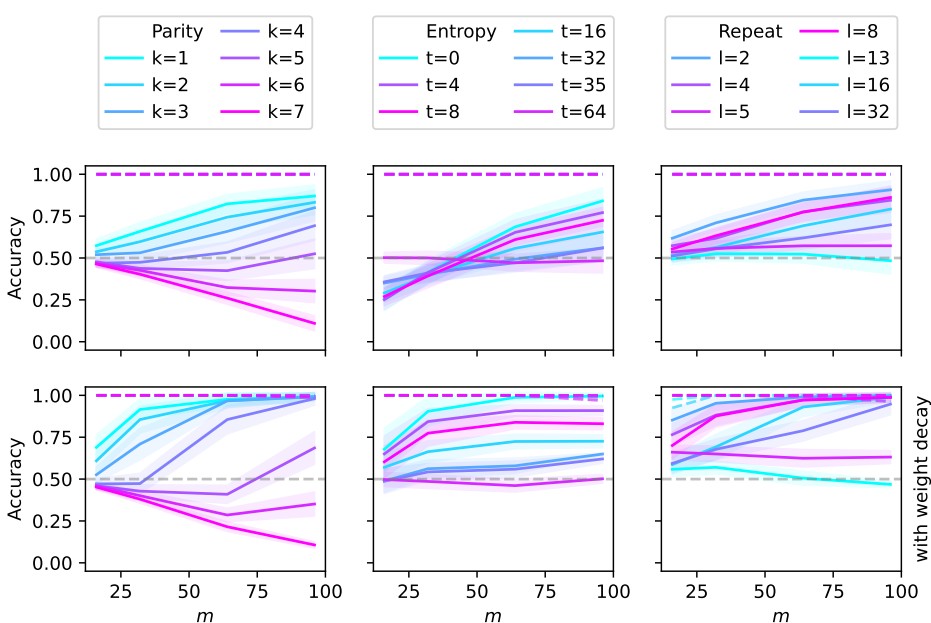

Figure 12: **SGD-like algorithm** trained on different targets from the $n = 7$ dataset. Each column shows a different function class – parity, entropy and repeat. See Appendix E.1 for full experimental details and a description of each function type. Comparing the top row (no weight decay) and the bottom row (weight decay; for this algorithm, $p = 0.3$) shows that weight decay massively outperforms non-weight decay on targets which have a small minimum representation (read: simple) and makes little difference for functions with a large minimum representation (read: complex functions). $l = 7$ parity is the most interesting example: it has the largest possible minimum representation and thus is the most complex function: the model is biased strongly against it, and thus the more data we give it, the worse it will be.

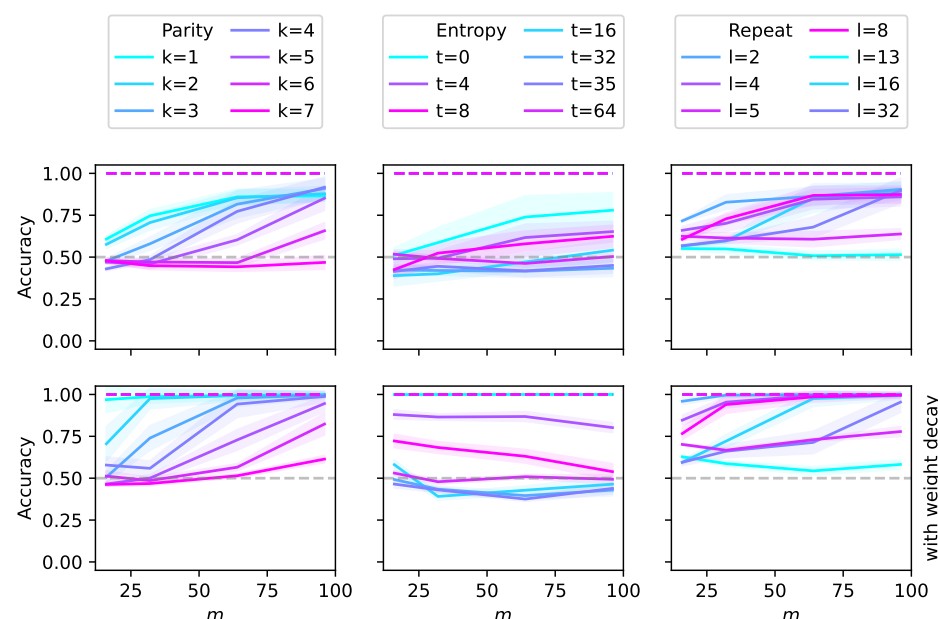

Figure 13: **SGD on continuous network** trained on different targets from the $n = 7$ dataset. Each column shows a different function class – parity, entropy and repeat. See Appendix E.1 for full experimental details and descriptions of each function type.

**Global vs. batch-wise local optimisation** The MCMC algorithm is a reversible Markov chain guaranteed to explore the full posterior. On the other hand, the steepest-descent algorithm never revisits rejected states – it hill climbs on randomly selected batch samples. The bias $p$ and norm-based tie-breaking act as heuristic "annealing" and "weight decay," but without detailed balance or stationary-distribution guarantees.

### E.6 CONTINUOUS NETWORKS

We also trained 2-layer FCNs with ReLU activations with the same width as the DFCNs. We used SGD with a batch size of $m/2$, and a fixed learning rate of $10^{-3}$. The results are shown in Figure 13. The only significant difference between the DFCNs trained with either Algorithm 1 or Algorithm 3 is large $k$-parity, which continuous neural networks have an easier time learning. This is because continuous networks exhibit slightly less inductive bias towards simple functions than DFCNs, which are manually crafted to fully take advantage of this bias for Boolean data.

