# OpenReview forum: "Characterising the Inductive Biases of Neural Networks on Boolean Data"
_ICLR.cc/2026/Conference — Submitted to ICLR 2026_

### Official Review · Reviewer_oevT · 2025-10-31

**Soundness:** 3
**Presentation:** 2
**Contribution:** 2
**Rating:** 2
**Confidence:** 3

**Summary:**

This works aims to derive a tractable framework towards understanding the inductive biases and generalization properties of neural networks. To this end, the authors consider Disjunctive Normal Forms (DNFs) and discrete depth two fully connected networks (DCNF) to analyze the aforementioned biases and properties in terms of appropriate complexity measures K(f) and prior probalities P(f). The correspondence shown in Proposition 2.7 allows mapping the networks weights (and architecture) directly in a symbolic logical representation, allowing investigations into the behavior of a function f with low DNF complexity, i.e., the shortest possible DNF expressing the function f, and the bias present in both untrained and trained architectures.

The main findings are that: 1) randomly initialized DFCNs induce a prior distribution P(f) that show a strong simplicity bias, i.e., simple functions occupy larger regions of the parameter space compared to complex ones, and 2) When training networks, simple functions generalize better and are easier to train due to the bias towards low complexity functions, leading at the same time to a difficulty in learning higher complexity functions.

**Strengths:**

Overall, the paper offers an interesting and principled analysis of a simplified theoretical model. The authors succeed in showing the connections between DNFs and DFCNs and how the function complexity leads to inductive biases and insights into networks generalization. This mapping is indeed insightful, and offers an alternative view towards representation learning.

**Weaknesses:**

My main concern revolves around the impact and generalization of the approach to other not so controlled settings. Indeed, while the authors aim to tie the generalization performance to derived inductive properties based on boolean function complexity, it remains highly uncertain how and if this discrete Boolean framework translates to more standard continuous, multi layer and high dimensional NNs.

Even if this work's aim is a proof of concept or a presentation of the connections between disjunctive normal forms and discrete fully connected networks, the focus on depth-two is itself very restrictive. I would have liked to see some insights or maybe derivations for networks with more than two layers, to assess how the proposed framework scales or adapts to different settings.

The authors apart from using depth two DFCNs, they also only consider ReLU activations. How would the framework behave if there was another activation in place (given that it satisfies the boolean nature of the examined setting)?

The main experiments are very limited. It is clear that the complexity of the proposed method is very high, leading the authors to consider only n=7. How are the wall time measurements as n increases?

The main text is dense, with many essential derivations and details deferred to the appendix. This renders the paper harder to follow and most of the time, breaks the reading flow. Moreover, some appropriate discussions that motivate and set the method in a specific context are also in the appendix.

**Questions:**

Please see the weaknesses section.

---

> ### Author Response · Authors · 2025-11-23
>
> **Weaknesses**
> Please also see our general comments for an overview of the paper’s aims and scope, which relate to your concerns around the impact and generalisation beyond our Boolean setting.
> - The restriction to depth-two networks is deliberate. Depth-2 DFCNs are exactly in bijection with DNFs (Proposition 2.7), which allows us to identify an architecture-coupled complexity measure K(f). These results can easily be extended to deeper networks (see [3]). Still, the two-layer setting demonstrates feature learning with interpretable features, and there was not enough room in the main text to add more content (given that it is already quite dense).
> - We would also like to address that our framework does not fundamentally rely on ReLU. To maintain the 1–1 correspondence between DFCNs and DNFs, we only need a monotone threshold-like nonlinearity so that each hidden unit behaves as a clause indicator. A hard step function or a (low-temperature) tanh activation would work equally well, with biases defined analogously to Definition 2.6.
> - On the question of larger n, see the general comments and the final appendix in the revised version where we treat larger n.
>
> [3] Mingard, Chris, et al. "Neural networks are a priori biased towards boolean functions with low entropy." arXiv preprint arXiv:1909.11522 (2019).

---

### Official Review · Reviewer_xwnS · 2025-11-01

**Soundness:** 4
**Presentation:** 3
**Contribution:** 3
**Rating:** 6
**Confidence:** 3

**Summary:**

The paper analyses the inductive bias of neural networks using a fully discrete, analytically tractable model.
A depth-2 discrete fully connected network (DFCN) with ternary weights $\{-1,0,1\}$ is shown to map one-to-one to Boolean DNF formulas, enabling a definition of function complexity $K(f)$.
By enumerating network configurations, the authors derive class-dependent asymptotic bounds for the prior $P(f)$: for example, t-entropy and k-parity functions follow distinct scaling laws (Table 2), showing that simple functions occupy much larger parameter volume.
Weight decay approximately adds an $e^{-\lambda K(f)}$ factor to the posterior, further amplifying this simplicity bias.
Small Boolean experiments qualitatively support the analytic results.

**Strengths:**

1. The paper presents a clear and internally consistent analysis. The proposed DFCN–DNF correspondence is an effective way to formalize inductive bias, allowing the authors to make explicit statements about which functions a network represents.
2. The results go beyond earlier simplicity-bias discussions by providing class-specific scaling laws (as shown in Table 2). This clarifies that different Boolean function families exhibit distinct probabilistic behavior rather than a single universal exponential law.
3. The connection between weight decay and a multiplicative $e^{-\lambda K(f)}$ factor in the posterior is well argued and conceptually coherent. The exposition is concise, and the figures and tables directly support the theoretical claims.

**Weaknesses:**

1. The work offers limited novelty. The simplicity bias of neural networks has been documented extensively, and this paper largely reformulates it within a discrete combinatorial model.
2. The theoretical depth is modest. the Boolean setting restricts the scope of the conclusions.
3. The MCMC and greedy search methods used for training differ substantially from gradient-based optimization, so their connection to real neural-network behavior remains unclear.
4. The presentation of related work could be better organized. Prior studies are frequently cited within the derivations themselves, which interrupts the flow and blurs the boundary between previous results and the authors’ own contributions.
5. The detailed enumeration over three hand-defined Boolean function families illustrates how the simplicity bias varies across structural types, but the result remains largely demonstrative rather than revealing new theoretical insight. Theorem 3.2 appears to bundle three independent asymptotic cases (constant, t-entropy, and k-parity) into a single statement mainly for narrative coherence.

**Questions:**

See Weaknesses.

ALso:
1. Could the authors move beyond the three enumerated Function Class and develop a more general or principled way to characterise function classes?

---

> ### Author Response · Authors · 2025-11-23
>
> We thank the reviewer for the careful reading and detailed comments.
>
> **Weakness 1**
> We agree that simplicity bias in neural networks is well-documented and did not intend to claim that we had somehow discovered it here. Nevertheless,  we do not feel that there remain important open questions, and in particular, that the causal mechanism is not sufficiently clear from past works. So we wanted to start from first principles: a complexity measure that is tightly coupled to the architecture, using this to explain the relation of $P(f)$ to $K(f)$, and then why this leads to generalisation.
>
> **Weakness 2**
>  With regard to the scope of the conclusions, we are simply aiming to demonstrate how some well-known phenomena (e.g. weight decay inducing feature learning) occur in a tractable model - hence using the function families by way of demonstration. See the global comment for more details, and for a more general point about the importance of toy models such as the one we study here.
>
> **Weakness 3**
> As we mentioned in the general comment, the posterior of our Metropolis–Hastings sampler (with a 0–1-like likelihood and weight norm penalty) has the same functional form as the posterior commonly of SGLD (e.g. [2]), when expressed in terms of weight norm and empirical loss. We will make this connection explicit when introducing Algorithm 1. In addition, our greedy SGD-like algorithm (Appendix E) shows qualitatively similar trends in learning curves across function families, which supports the claim that the posterior picture is relevant for SGD in this discrete setting.
>
> **Weakness 4**
> With regards to the presentation, we found it hard to squeeze in all the background and new results into 9 pages (an all too common issue), and that led to compromises.  If the reviewer has any particular sections they found unclear we can attempt to rewrite.
>
> [2] Naveh, Gadi, et al. "Predicting the outputs of finite deep neural networks trained with noisy gradients." Physical Review E 104.6 (2021): 064301.

---

### Official Review · Reviewer_sVta · 2025-11-08

**Soundness:** 3
**Presentation:** 3
**Contribution:** 2
**Rating:** 4
**Confidence:** 3

**Summary:**

This paper introduces a framework to study inductive bias, feature learning, and generalization in neural networks by focusing on depth-2 discrete fully connected networks (DFCNs) trained on Boolean functions. They show a bijective correspondence between DFCNs and disjunctive normal form (DNF) formulas. This equivalence allows the authors to define a function-level complexity measure K(f) based on the minimal DNF length and to link it directly to network weight norms.
They empirically and theoretically derive a simplicity-biased prior P(f) over Boolean functions, showing that functions with small K(f) occupy exponentially larger regions of parameter space. Using Bayesian and greedy SGD-like training algorithms, the paper demonstrates that generalization correlates strongly with DNF complexity. They further show that weight decay improves generalization for low-complexity targets.

**Strengths:**

1.	Novel framework for understanding inductive bias of NNs.
2.	New insights relating DNF complexity and learnability.
3.	The paper is mostly clearly written.
4.	Thorough empirical validation.

**Weaknesses:**

1. The main weakness is that the framework limits the input dimension substantially, allowing only n≤7. Therefore, it does not model high dimensional settings, which are key in NN applications. Since the data is binary, the resulting datasets are also very small.
2. No experiments were performed with NNs and algorithms used in practice – this can strengthen the conclusions of the paper.
3. Missing reference– Bronstein et al. (UAI 2022) that study the inductive bias of NNs on read-once DNFs.


Bronstein, Ido, Alon Brutzkus, and Amir Globerson. "On the inductive bias of neural networks for learning read-once dnfs." Uncertainty in Artificial Intelligence. PMLR, 2022.

**Questions:**

Lines 244-247: the PAC-Bayes bound is unclear. How was the KL divergence derived? What exactly is ϵ(f)? Does the right-hand side approach 1 when P(f) is large?

In the experiments, what is the main factor that limits the input dimension? Are there ways to relax this restriction and show that the conclusions hold (even approximately) for higher dimensional settings?

---

> ### Author Response · Authors · 2025-11-23
>
> We thank the reviewer for the careful reading and detailed comments.
>
> **Weaknesses 1 and 2**:  Our general comment covers the general scope of the paper and small $n$
>
> **Weakness 3**: We appreciate the reviewer’s helpful pointer to Bronstein et al. (UAI 2022), which we will add and discuss explicitly. While their work analyses the inductive bias of gradient flow on convex one-hidden-layer ReLU networks and shows that min-norm KKT solutions correspond to read-once DNF-aligned networks, we think our DFCN framework complements this prior work by providing a discrete model in which the feature space is exactly the space of DNF clauses - explaining training away from the minimum norm solution. We explain the same preference for DNF-aligned solutions in terms of prior volume and posterior bias, while also quantifying when this mechanism fails (e.g., high-order parity).

---

### Author Response · Authors · 2025-11-23
**General comments**

We appreciate the reviewers’ time and feedback. We suspect the core aim of the paper may not have come through clearly. Our goal is to present a toy setting that is just rich enough to exhibit the basic mechanisms behind generalisation in over‑parameterised networks, yet small enough that one can analyse the entire causal chain step by step. . This includes how an architecture‑induced prior over functions leads to certain observed training dynamics and feature learning, and finally test performance. We believe such an end-to-end explanation is unique in the literature and has independent value.

More generally, **there is always a trade-off between tractable but simplified models and complex models whose learning behaviour cannot yet be fully captured analytically**. Most submissions to ICLR focus on more complex neural networks. Our work belongs to a smaller set of papers that employ simplified, analytically tractable toy models to achieve deep understanding, even if the setting is limited. While simplified models can be critiqued for lack of realism, and complex models for their intractability, progress requires exploring both types of approaches.

Concretely, our setting offers four ingredients that, to our knowledge, have not previously been shown together in one tractable framework:

1. Full visibility of the function space. Because the task is discrete, we can reason about and, at small n, evaluate on the entire hypercube. This makes the no-free-lunch trade‑offs concrete - e.g. high‑order parity sits in a vanishing‑mass region of the prior, so adding additional data to the training pool does not help and can even harm generalisation. Furthermore, because parity is the function with max possible complexity/norm, feature learning actively hurts generalisation.
2. A one‑to‑one DFCN–DNF correspondence. Our depth‑2 discrete FCN (DFCN) admits a bijection to DNFs. This yields an architecture‑coupled complexity K(f) which ties the minimal DNF length directly to weight norms. The bijection lets us mechanistically explain the inductive bias by identifying which Boolean functions occupy large parameter‑space volume, and why. This also clarifies where “volume arguments” from e.g. [1] come from and why low‑K functions tend to have high prior probability (with notable exceptions like parity).
3. Interpretable feature learning. Weight decay translates into a simple posterior multiplier (e^{-\lambda ||\theta||^2}) (Eq. 7), sharpening the native simplicity bias and selecting low‑complexity DNFs. In this model the features are explicit clauses, so it is clear what is learned and why those features arise under training. This is a rare case where we can show feature learning improving generalisation on easy targets and failing on inherently complex ones (e.g., high‑order parity). Furthermore, it is clear that weight decay is actively penalising DNF complexity.
4. MCMC vs SGLD. Our sampler induces a posterior that has the same functional form as the posterior induced by SGLD [2], making the link to gradient‑based training explicit. We will emphasise this clearly in the paper.

We note a few examples
- Mechanistic interpretability shows what emerges (e.g., modular arithmetic circuits or polysemantic heads) but, by design, often fails to explain why that ties emergence to architecture‑level priors and posterior volume. Our framework provides that missing causal pathway in a setting where features are transparent.
- Kernel/NTK analyses (e.g., Canatar et al., 2021) quantify generalisation without feature learning, but do not provide a mechanistic explanation for why kernels have a decaying eigenspectrum, or why the large eigenmodes are useful functions. Furthermore, our DFCN admits feature learning and lets us say when it helps or hurts, using an interpretable complexity K(f) and the induced P(f).
- Gradient‑flow/KKT results for read‑once DNFs (Bronstein et al., UAI’22) prove that GF avoids memorisation and converges to min‑norm, term‑aligned solutions. We tell a complementary story, namely the prior‑and‑posterior picture that explains why those aligned minima are preferred and when that preference is insufficient (e.g., parity).

If the reviewers feel that this needs to be better explained, we would be happy to rewrite the introduction and discussion to emphasise these points more effectively.

One issue the reviewers brought up was the dataset size. We want to make clear that the aim of the paper is not to claim scale‑robust performance. We chose n=7 so that we could evaluate all test points and use a network that guarantees full expressivity (first‑layer width (2^{n-1})). However, to demonstrate that our results do apply at larger n, we will add a few plots with larger n in the final Appendix when we upload a revised submission.

(Comment broken up into two parts due to character constraints, 1 of 2)

---

> ### Author Response · Authors · 2025-11-23
>
> In summary, this work is intentionally a small, exactly‑solvable model that (1) ties architecture-prior mass P(f), (2) yields a human‑readable complexity K(f) tied to weights, (3) shows feature learning in a setting where the features are unambiguous, and (4) explains when weight decay helps, and when no amount of data helps (e.g., high‑order parity). These achievements would be very hard with a larger, more complex model.  We consider our contributions as a useful guide to what may be potentially missing building blocks for a full  theory of generalisation.
>
> [1] Valle-Perez, Guillermo, Chico Q. Camargo, and Ard A. Louis. "Deep learning generalizes because the parameter-function map is biased towards simple functions." arXiv preprint arXiv:1805.08522 (2018).
> [2] Naveh, Gadi, et al. "Predicting the outputs of finite deep neural networks trained with noisy gradients." Physical Review E 104.6 (2021): 064301.

---

### Meta-Review · Area_Chair_pvsi · 2025-12-19

**Summary:**

Reviewers all commented that the paper is rigorous and theoretically well-motivated, yet unanimously agree (with different weighting in their scores) that the scope is rather limited. An additional concern was that the phenomena discussed in the paper are at least in parts well documented in existing prior work already. Overall, the paper seems to make a deliberate choice that favors simplicity and allows for theoretical derivations, but this seems to come at the expense of a partial disconnect to practical settings.

**Reviewer Concerns:**

The individual rebuttals mostly consist of concise clarifications of a deliberate limitation in scope of the paper and conscious choices made to enable the conducted analysis. As such, they primarily aim to clarify, rather than suggest concrete solutions to the reviewers concerns. As such, the AC believes that most of the reviewers concerns are left unaddressed.

The only exception is visible from the general response of the authors, where on top of clarifying the scope, other related works are further mentioned to stress the potential value of papers that focus on analytical settings (in trade-off to complex settings). The authors mention that a discussion could be added, and that introduction and related work could be rewritten accordingly to make this point clear.

**Reviewer Scores:**

Reviewer scores ranged from reject (2) to marginally below acceptance (4) to marginally above the acceptance threshold (6). Although the scores differ, the reviewers largely agree wrt their concerns in terms of scope, depth, and applicability. Given that the rebuttal primarily aims to clarify the scope rather than extend the provided analysis, the AC believes it is highly unlikely that the scores would have changed substantially. Although there was no possibility for a discussion due to the incident, not much content has been provided in the rebuttal as a respective basis.

---

### Decision · Program_Chairs · 2026-01-26

Reject